# Dystonin modifiers of junctional epidermolysis bullosa and models of epidermolysis bullosa simplex without dystonia musculorum

**Thomas J. Sproule**[1]*, **Robert Y. Wilpan**[1], **John J. Wilson**[1], **Benjamin E. Low**[1], **Yudai Kabata**[2,3], **Tatsuo Ushiki**[3], **Riichiro Abe**[2], **Michael V. Wiles**[1], **Derry C. Roopenian**[1], **John P. Sundberg**[1,4]

1 The Jackson Laboratory, Bar Harbor, ME, United States of America, 2 Division of Dermatology, Graduate School of Medical and Dental Sciences, Niigata University, Niigata, Japan, 3 Division of Microscopic Anatomy, Graduate School of Medical and Dental Sciences, Niigata University, Niigata, Japan, 4 Department of Dermatology, Vanderbilt University Medical Center, Nashville, TN, United States of America

* tom.sproule@jax.org

**Data Availability Statement:** All relevant data are within the manuscript and its Supporting Information files.

## Abstract

The *Lamc2^jeb* junctional epidermolysis bullosa (EB) mouse model has been used to demonstrate that significant genetic modification of EB symptoms is possible, identifying as modifiers *Col17a1* and six other quantitative trait loci, several with strong candidate genes including dystonin (*Dst/Bpag1*). Here, CRISPR/Cas9 was used to alter exon 23 in mouse skin specific isoform *Dst-e* (Ensembl GRCm38 transcript name Dst-213, transcript ID ENSMUST00000183302.5, protein size 2639AA) and validate a proposed arginine/glutamine difference at amino acid p1226 in B6 versus 129 mice as a modifier of EB. Frame shift deletions (FSD) in mouse *Dst-e* exon 23 (*Dst-e^FSD/FSD*) were also identified that cause mice carrying wild-type *Lamc2* to develop a phenotype similar to human EB simplex without dystonia musculorum. When combined, *Dst-e^FSD/FSD* modifies *Lamc2^jeb/jeb* (FSD+*jeb*) induced disease in unexpected ways implicating an altered balance between DST-e (BPAG1e) and a rarely reported rodless DST-eS (BPAG1eS) in epithelium as a possible mechanism. Further, FSD+*jeb* mice with pinnae removed are found to provide a test bed for studying internal epithelium EB disease and treatment without severe skin disease as a limiting factor while also revealing and accelerating significant nasopharynx symptoms present but not previously noted in *Lamc2^jeb/jeb* mice.

## Introduction

Epidermolysis bullosa (EB) is a group of rare disorders in humans caused by mutations in any of at least 20 genes resulting in weakening of the dermal-epithelial connection leading to blistering with a range in severity from neonatal lethal to mild adult onset [1]. EB has largely been treated as a group of monogenic simple Mendelian diseases [2, 3]. Evidence of genetic

**Funding:** DCR received award Roopenian-1 from Debra Austria https://www.debra-austria.org/ and Roopenian-2 from Debra UK https://www.debra.org.uk/. The funders had no role in study design, data collection and analysis, decision to publish, or preparation of the manuscript.

**Competing interests:** The authors have declared that no competing interests exist.

modification of symptoms has been building, but rarity of individual alleles and complexities of population genetics make identification of modifier genes in humans difficult [4]. While one human study identified *MMP1* as a genetic modifier of a single *COL7A1* mutation that caused dystrophic EB [5], this finding was not confirmed in two follow up studies using different patient cohorts and *COL7A1* mutations [6, 7]. To date, there have been few suggested modifiers EB in humans or animal models [8–12]. The *Lamc2^{jeb/jeb}* spontaneous hypomorphic mouse model of intermediate junctional EB (JEB) [13] has been used to successfully identify EB modifier genetics in mice. This hypomorphic model was used recently to map seven unique quantitative trait loci (QTL) that modify the *Lamc2^{jeb/jeb}* syndrome. One such strong modifier was formally identified to be caused by 2–3 naturally occurring, normally innocuous missense nucleotide polymorphisms (SNPs) mapping to the non-collagenous 4 domain of collagen XVII (COLXVII) encoded by the 'EB related gene' *Col17a1* [14].

This paper investigates a second strong QTL that arose from the *Lamc2^{jeb/jeb}* modifier mapping studies. This QTL localized to mouse chr1:24-38Mb (peak 34Mb) in a C57BL/6J (B6) x 129X1/SvJ (129X1) $F_2$ (B6129X1$F_2$) cross. It includes another 'EB related gene', dystonin (*Dst*) [14]. Dystonin, previously identified as BPAG1 and BP230 in studies of the autoimmune disease bullous pemphigoid [15, 16], was later determined to be a large gene with complex transcripts in both mouse and human, including highly nerve, muscle and skin (epidermal) specific isoforms initially dubbed *BPAG1a*, *BPAG1b* and *BPAG1e* respectively and later named *Dst-a*, *Dst-b* and *Dst-e* [17–20]. The nerve and muscle isoforms share the majority of the ~100 *Dst* exons, differing from each other by only a few. *Dst-e* (Ensembl GRCm38 transcript name Dst-213, transcript ID ENSMUST00000183302.5, protein size 2639AA) contains only 24 exons, of which 21 (exons 2–22), encoding ~1/3 of the protein, are shared with *Dst-a* and *Dst-b*. DST-e is found mainly in hemidesmosomes of epithelial basal keratinocytes but to a lesser level in cornea, urinary bladder, stomach and large intestine (biogps.org). Its homodimeric structure consists of 3 domains: a plakin domain that binds transmembrane collagen XVII (COL17A1) and integrin α6β4 (ITGA6, ITGB4), central coiled-coil rod and intermediate filament binding domain (IFBD1) which binds intercellular keratin 5/14 (KRT5, KRT14) dimers. The rod and IFBD1, coded by *Dst-e* exons 23 and 24 respectively, are likely exclusive to the epithelial isoform, though a peripheral nerve specific isoform *BPAG1-n*/*Dst-n* containing them has been proposed [18, 21]. Deleterious mutations in *Dst-a* exons, including those shared with *Dst-e*, have severe consequences due to the critical nature of the central nervous system isoform of this protein, resulting in dystonia musculorum (DM) in mice and hereditary sensory and autonomic neuropathy type VI (HSAN6) in humans [22–25]. EB is likely in patients and mouse models when mutations affect exons coding both *Dst-a* and *Dst-e*, though it can easily be overshadowed by the typically much more severe nervous system disorders. EB has been reported in some HSAN6 cases [22, 26] and not others [23, 27] in which shared exons are affected. A few documented cases of humans with frame shift deletion mutations in *DST-e* exon 23 have exhibited mild EB simplex symptoms with mild or no nervous system involvement [28–31], as would be predicted since exon 23 is not included in nerve specific isoform *DST-a*. A muscle specific *Dst-b* knockout has recently been generated in mice [20]. No human cases or mouse models disrupting only *Dst-e* exons 1 or 24 have yet been reported.

For this study, CRISPR/Cas9 was used to modify *Dst-e* specific exon 23 in B6 mice. Based on previous results with *Col17a1*, 1–2 B6/129X1 missense polymorphisms in *Dst-e* exon 23 resulting in AA p1226 R/Q and p1469 G/V were hypothesized to be responsible for the observed phenotypic differences [14]. A successful replacement by homologous directed repair (HDR) and in-frame deletions (IFD) at the first locus, and frame shift deletions (FSD) affecting both loci were created and tested for standalone phenotypes in B6 mice and as modifiers of JEB in *Lamc2^{jeb/jeb}* mice. The amino acid (AA) 1226R→Q replacement partially recapitulated

the 129 'more protective' phenotype, confirming its involvement while also implicating contribution of the second missense change (1469G→V) and/or other mutations within or adjoining dystonin to the modifier effect. IFD and FSD also alter B6-*Lamc2*$^{jeb/jeb}$ phenotypes, confirming and expanding information about *Dst-e* coiled-coil mutation modifier effects. FSD additionally result in new *Dst* EB simplex (EBS) mouse models without the complication of dystonia musculorum (DM) present in all previous models, allowing the ability to study phenotypes much later in life. Differences in EBS symptoms between FSD and previous models which included DM, together with unanticipated phenotypes in FSD mice with *Lamc2*$^{jeb/jeb}$ (FSD+*jeb*), provide evidence for a second rodless epithelial dystonin, *Dst-eS/Bpag1eS* [32], which differs in function from *Dst-e*.

## Results

### CRISPR/Cas9 founders

CRISPR/Cas9 mediated genetic modification was used to determine if missense mutations in the epithelial form of the 'EB related gene' dystonin (*Dst-e* or *Bpag1e*, Ensembl mouse transcript Dst-213) were responsible for the observed chr1:24-38Mb B6/129X1 modifier effect [14]. Per the Sanger Mouse Genome Project REL-1410 in conjunction with Ensembl and Mouse Phenome Database (MPD) SNP/genotype variation (later replaced by GenomeMUSter Search (mpd.jax.org/genotypes)), there are 26 missense SNPs between B6 and 129S1 in *Dst*, 129S1 and 129X1 share strain heritage at *Dst* so Sanger information could be applied equally to both, and only 5 of those map to the epithelial form *Dst-e*/Dst-213 [18]. B6.A chr1 congenic results reduce the *Dst-e* candidate missense list to only the two at which B6 and A have the same allele: p1226 R/Q and p1469 G/V, located in *Dst-e* proximal and mid exon 23 respectively (**Fig 1A and 1B**) [14]. These were separately targeted for CRISPR/Cas9 double stranded cutting and SNP specific replacement by homologous directed repair (HDR) in B6 zygotes. While both attempts produced numerous targeted deletion founders, demonstrating successful CRISPR cutting, the first target produced only one successful SNP replacement founder (**Fig 1C**) and the second none. The replacement founder (called *em1*, based on assigned allele name *Dst*$^{em1Dcr}$) and select in-frame (*em2-5*) and frame shift (*em6-16*) deletion founders were bred to be separately on both B6 wild-type and B6-*Lamc2*$^{jeb/jeb}$ backgrounds. All mutant strains were bred to homozygosity and characterized, primarily by tail tension test at 10 weeks of age and extended aging for ear, tail and body condition scores (**Figs 2 and 3**).

### *Dst-e em* mutation effects on a B6 (*Lamc2*$^{wt/wt}$) background

The *em1 Dst-e* AA 1226R→Q replacement as well as selected small and large in-frame deletions (IFD) and frame shift deletions (FSD) of 7–1016 bp, all entirely within *Dst-e* exon 23 (**Fig 4**), were made homozygous and tested on a B6 background (*Lamc2*$^{wt/wt}$) by observation and ear, tail and trunk skin condition 'scoring' up to one year of age. Additional assays included male tail tension tests at 10 and 20 weeks of age, qPCR from tail skin cDNA to confirm dysregulation of *Dst-e* and capillary immunoelectrophoresis from neonatal trunk skin. The *em1* replacement and all IFD did not alter phenotypes in any way tested (**Figs 2A and 5–9**) while all FSD resulted in trunk hair loss beginning at 8–10 weeks of age proceeding to blisters in some after 12 (*em13*) to 18 (*em14*) weeks of age, sometimes accompanied by ear ulcers (**Fig 2**). FSD tail tension tests failed to remove skin 'sleeves' in all but one male tested at 10 or 20 weeks old or in two *em14* tested at 54 weeks old, like IFD and B6 negative controls and unlike B6-*Lamc2*$^{jeb/jeb}$ positive controls, though there was a suggestive mild increase in skin removal by tail tension test compared to B6 controls (**Figs 2G and 2H and 7**). FSD tails did not develop lesions or other imperfections by 54 weeks of age, the oldest age observed (**Fig 2B and 2F**).

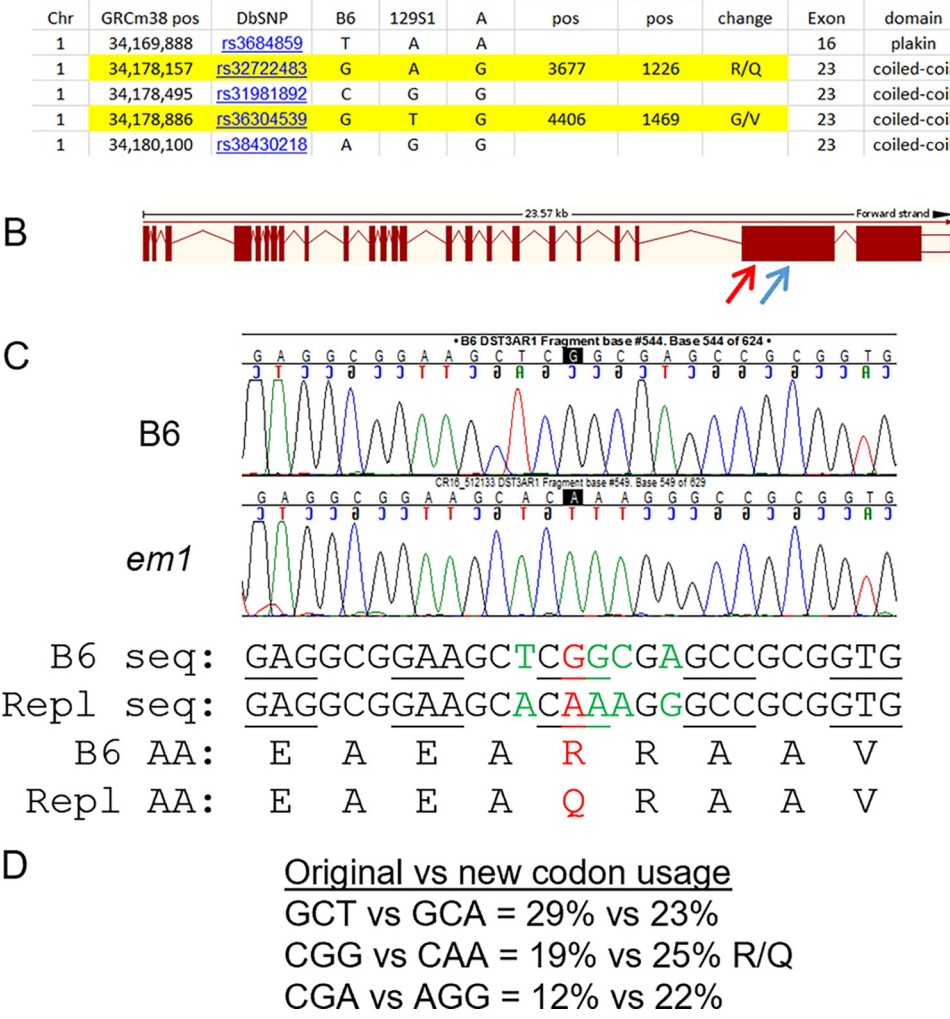

Fig 1. CRISPR/Cas9 amino acid change. (A) A list of B6 vs 129 vs A mouse missense SNPs mapping to *DST-e*, showing that 129 and A differ at 2 of 5. (B) Mouse DST-213 exon/intron placement and spacing from Ensembl with indicated CRISPR/Cas9 cut sites in exon 23 Dst3A (red arrow) and Dst3B (blue arrow). (C) Sequence showing successful Dst3A five nucleotide replacement resulting in one amino acid change from B6 allele to 129 allele: p1226 R→Q. Sequence shown is mouse chr1: 34,178,144–34,178,170 (GRCm38). (D) Codon usage of original and replacement amino acids.

FSD lines tested differed statistically in ear and body percentages affected and ages of onset as measured by age at which each reached a severity score of '4' (moderate, **Fig 2B**).

By light microscopy, FSD ears and tails appeared normal at 6 and 16 weeks of age but by 30–35 weeks old most demonstrated ear pinnae with dermal inflammation, edema and epidermal thickening (7 of 9 mice) as well as small areas of ear and tail dermal-epidermal separation (7 of 9 and 4 of 9 respectively) not seen in any of the B6 or IFD mice (**Fig 2I–2O**). FSD ears included regions of both epidermal and dermal thickening (asterisk and boxed area respectively in **Fig 2J**). The dermal changes were due to dilation of lymphatics (white arrowheads, **Fig 2K**) and a mixed inflammatory cell infiltrate (black arrowheads, **Fig 2K**) including lymphocytes, neutrophils, macrophages, melanomacrophages, and fibroblasts that separated the dense irregular collagenous connective tissue. Additionally dorsal skin, footpads, ears and tails

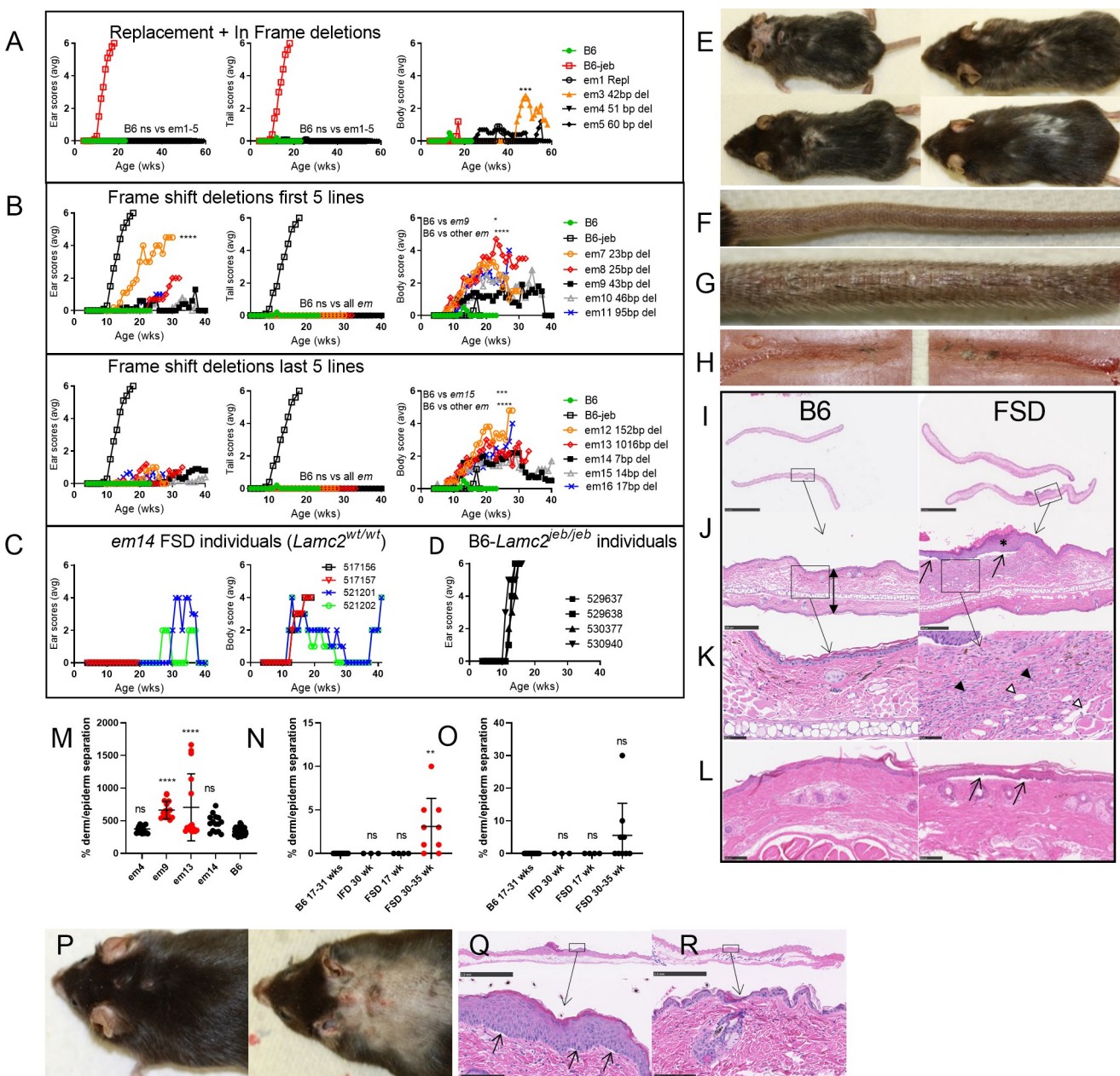

**Fig 2. *Dst-e* exon 23 mutation results on B6 (*Lamc2^{wt/wt}*) background.** (A) *em1* 1226R→Q replacement and *em3-5* IFD ear, tail and body scores compared to B6 and B6-*Lamc2^{jeb/jeb}* (B6-jeb) controls. Ear, tail and body score statistics 1-way ANOVA based on age at which scores reached '4' (moderately affected) for each mouse comparing *em* lines to B6 controls: ear scores all ns (>0.05); tail scores all ns; body score B6 vs *em3* p<0.001, ns vs all others. (B) FSD ear, tail and body scores split to two sets of graphs for clarity (5 *em* lines per graph) with same controls and statistics as (A): ear scores B6 vs *em7* p<0.0001, ns vs all others; tail scores B6 ns vs all; body scores B6 vs *em9* p<0.05, B6 vs *em15* p<0.001, B6 vs all others p<0.0001. (A-B) statistics for pair wise comparisons of all lines are included in supporting information. (C) Ear and body scores from typical individual Dst *em14* FSD mice demonstrating recovery once affected. (D) Individual ear scores from typical B6-*Lamc2^{jeb/jeb}* demonstrating they do not recover. (E) Examples of FSD ears, neck and trunk developing lesions while (F) tail unaffected. (G-H) Gross results of FSD tension test (compared to controls in **Fig 8**). *Em13* FSD 20 week old males shown in (E-H) are typical of extent of disease per mouse across frame shift lines, except for percent affected and time of onset differences noted in (B). (I-L) Typical B6 and *em9* FSD ear (I, magnified J and K) and tail (L) H&E examples from a set of *em9* FSD at 30–35 weeks old and B6 controls at 26 and 31 weeks old. Bar is 2.5 mm (I), 250 μm (J), 50 μm (K) and 100 μm (L). Arrows indicate dermal-epidermal separation. White arrowheads indicate dilation of lymphatics. Black arrowheads indicate cell infiltrates. Asterisk denotes epidermal thickening. (M) Ear thickness of 26–35 week old samples in (I) in μm, top of stratum corneum one surface to top of stratum corneum on opposite surface in areas with no tearing or artifacts (i.e. double-headed arrow in J) based on five measurements per ear at middle. (N-O) Graphical representation of the amount of dermal-epidermal separation in (I) H&E slides. (M-O) Error bars are SD. P-values are compared to B6. **** p<0.0001, ** p<0.01, ns is not significant. (P) Photos of a 44 week old male *em14* FSD before and after shaving (ears were removed for histology). (Q-R) Dorsal trunk skin H&E from 44 week old *em14* FSD male with lesions (Q) and 54 week old *em4* IFD

male without (R). Bar is 2.5 mm top and 100 μm bottom. All results *Lamc2^{wt/wt}* except (D) and B6-jeb is B6-*Lamc2^{jeb/jeb}* controls in (A-B). B6 and B6-jeb controls are the same in (A) and (B).

were examined histologically from 44-week-old *em14* FSD and 54-week-old *em4* IFD as controls. Shaving of the backs of these mice revealed the extent of blistering in symptomatic mice (**Fig 2P**). Dorsal trunk skin exhibited epidermal thickening in regions where lesions had been observed (2 of 4 FSD) and small regions of dermal-epidermal separation (3 of 4 FSD, ~5% of total boundary) in both areas with normal or thickened epidermis (**Fig 2Q**). No such separations were noted in IFD (**Fig 2R**). No obvious skin abnormalities were noted in footpads or nail units of IFD or FSD like those found in *Lamc2^{jeb/jeb} Col17a1^{em1Dcr/em1Dcr}* mice [14].

Capillary immunoelectrophoresis of neonatal trunk skin using a polyclonal anti-dystonin antibody gave 6 peaks in the range 2–440 kDa, of which the single largest ~247 kDa was consistently absent from FSD samples (**Fig 5A and 5B**). Immunohistochemical (IHC) localization of nerves using an antibody directed against ubiquitin carboxy-terminal hydrolase L1 (UCHL1, also called PGP9.5) revealed that there was no difference in the number, distribution or anatomical structure of nerves in the skin between *em13* FSD, *em14* FSD and B6 controls (**Fig 5C**), indicating that peripheral nerves contain *Dst-a* derived protein, which skips the exon mutated in FSD, rather than proposed *Dst-n* which includes it, and should be functionally null in these mice [18].

Transmission electron microscopy (TEM) of dorsal trunk skin and footpad revealed an absence of the hemidesmosomal inner plaque but no other abnormalities in *em13* and *em14* FSD compared to B6 controls (**Fig 6**). 'Filaments' are observed apparently attached to HD outer plaques in some images, but we were unable to confirm that they were keratin intermediate filaments (IF), pieces of the outer plaque or other structures. *Em13* and *em14* TEM was compared to *Dst^{dt-23Rbrc}* homozygous mice previously reported [19], which disrupts *Dst-e* exon 14 (*Dst-a* exon 27), by an author associated with both projects. No clear differences were observed.

A check for sex linked disease dimorphism was made by comparing body scores in *em14* FSD mice. Age when body score first reached '4' (moderately affected) revealed a trend of females experiencing less disease than males, similar to that observed for *Lamc2^{jeb/jeb}* on multiple strain backgrounds [33], but it did not reach statistical significance (**Fig 8F**).

## cDNA analysis/*Dst-eS* presence

Tail skin cDNA from all lines (replacement, IFD and FSD) was tested using PCR with primers from *Dst-e/Dst-213* exon 22 to 23 and within exon 23 to determine transcripts made in *Dst^{em}* lines, especially the frame shift mutations (**Table 1**). Primer pair Exon22F-Exon24R were also tested to determine if FSD resulted in an exon skipping transcript. Exon 22–23 and exon 23 internal primers all produced bands except when primers mapped within the deleted region (*em13*), suggesting that RNA transcription was not disrupted by the deletions, including frame shifts. Product sizes shifted as predicted (**Fig 7B**). Exon 22F-24R PCR gave products for all, including IFD and the B6 control. *Dst-eS/Bpag1eS* is the only published dystonin isoform expected to give this result [32]. Exon 22F-24R cDNA PCR products were sequenced and confirmed to read directly from the end of Dst-213 exon 22 into the beginning of exon 24 (**Fig 7C**). Tail skin cDNA from all lines were qPCR tested for *Dst-e* regions, including 22F-24R. *Em* line Ct indicating RNA quantity generally matched the B6 control, though FSD may have averaged 2–4 fold less for most markers (**Fig 7D**). To address the possibility of tissue specific differential transcripts in FSD+*jeb* mice, ear and tail were separately collected from two B6 mice. Samples were qPCR tested using cDNA with *Dst-e* exon 22–23, 23–23, 23–24 and 22–24

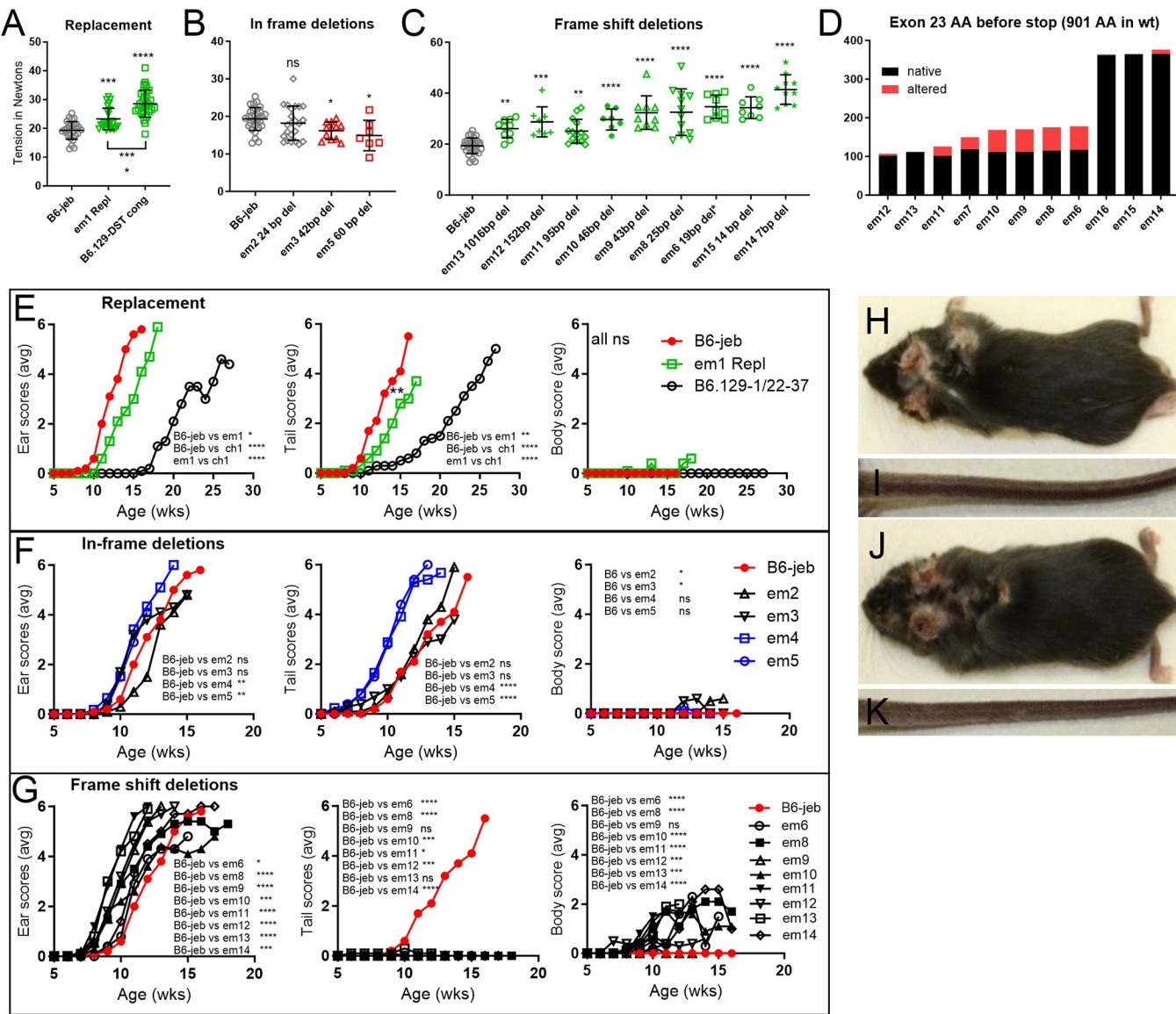

**Fig 3. *Dst-e* exon 23 mutation results on B6-*Lamc2^jeb/jeb^* background.** (A-C) Tension results for (A) *em1* 1226 R→Q replacement compared to the B6.129 DST congenic it was trying to explain, (B) in frame deletions (IFD) and (C) frame shift deletions (FSD). (C) ordered by deletion size in decreasing order. (D) FSD number of amino acids coded in exon 23 before stop codon, black indicates native, red altered. (E-G) Ear scores, tail scores and body scores for same as (A-C): (E) replacement, (F) in-frame deletions and (G) frame shift deletions (B6 = B6-*Lamc2^jeb/jeb^*). (H-K) Examples of affected ears and bodies but unaffected or minimally affected tails from 2 FSD strains: *em12* (H,I) and *em13* (J,K) at 10 weeks of age. Tension data are all at 10 weeks of age. Tension error bars are SD. Statistics above tension data are vs B6-*Lamc2^jeb/jeb^* control, below compare indicated groups, p: ns >0.05, * <0.05, ** <0.01, *** <0.001, **** <0.0001.

primers. Tail had ~32 fold lower expression using exon 22–24 primers, representing *Dst-eS*, than exon 22–23, 23–23 and 23–24 primer pairs representing *Dst-e*. Ear and tail had similar expression levels to each other except ear was ~4 fold less than tail using exon 22–24 primers (**Fig 7E**). To compare mouse results to human and confirm published results [32], human adult skin cDNA was similarly PCR tested with *DST-e* exon 22–23, 23–23, 23–24 and 22–24 primer pairs (**Table 1**) and exon 22–24 PCR product was sequenced, confirming the presence of a comparable exon 23 skipping product in humans (**Fig 7F and 7G**).

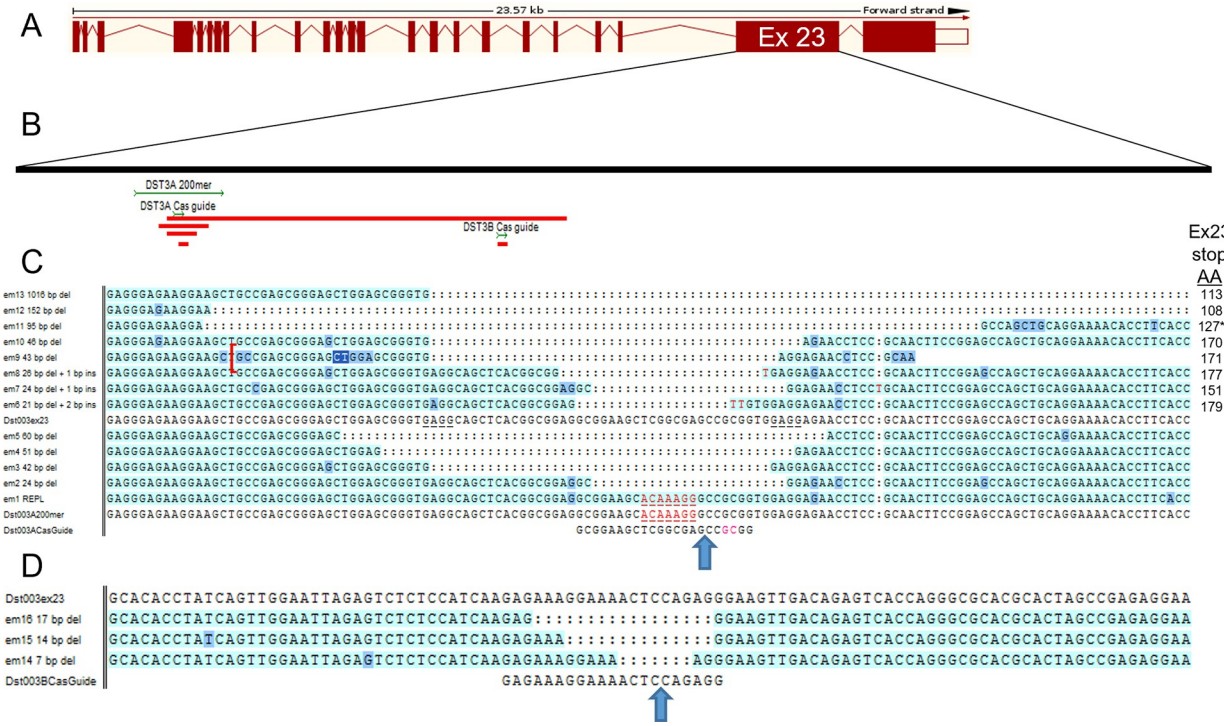

**Fig 4. Sequence of induced mutations.** (A) Mouse *Dst-e* exon-intron format from Ensembl. (B) Expanded exon 23 showing approximate positions of Dst3A and Dst3B Cas9 guides and cut sites, Dst3A 200mer for replacement and example deletions (red). (C) Sequence alignment of DNA deletions and replacement at Dst3A cut site as analyzed in Sequencher software (frame-shift deletions top, in-frame deletions next, replacement *em1* bottom). (D) Sequence alignment of Dst3B deletions–all frame shifts.

### *Dst-e em* mutation effects on B6-*Lamc2*<sup>jeb/jeb</sup> background

The *em1* homozygous mutation on a B6-*Lamc2*<sup>jeb/jeb</sup> background was significantly protective by tension test, ear scores and tail scores compared to B6-*Lamc2*<sup>jeb/jeb</sup> controls but was far from accounting for the full B6.129 congenic modifier effect, demonstrating that AA p1226 missense changes do modify *Lamc2*<sup>jeb/jeb</sup> disease but also indicating that p1469 G/V or other polymorphisms also play a role (**Fig 3A and 3E**). Because only one mouse was tested from each line in the trunk skin protein assay it cannot be ruled out that B6/129 expression differences might account for some portion of the *Lamc2*<sup>jeb/jeb</sup> disease onset differential (**Fig 5A and 5B**).

IFD and FSD lines were also made homozygous on a B6-*Lamc2*<sup>jeb/jeb</sup> background (henceforth referred to as IFD+*jeb* and FSD+*jeb*). Most, if not all, *Dst-e* exon 23 deletion mutants altered the B6-*Lamc2*<sup>jeb/jeb</sup> phenotype in different ways, confirming that *Dst-e* rod alterations can serve to modify laminin 332 (L332) connectivity and JEB disease in *Lamc2*<sup>jeb/jeb</sup> mice. IFD +*jeb*, removing 24–60 bp of the *Dst-e* proximal coiled-coil rod in-frame all suggested a trend toward increasingly detrimental phenotypes with increased deletion size (**Fig 3B and 3F**). FSD +*jeb*, in which the mutations are predicted to cause a *Dst-e* specific premature stop codon and likely nonsense mediated decay and a functionally null state, displayed an unexpected split phenotype. In all previous mouse *Lamc2*<sup>jeb/jeb</sup> experiments, all scoring parameters agreed, with one strain being 'more protective' or 'more detrimental' than another by all three standard measurements: tail tension test, ear scores and tail scores, although not always with comparable significance [14, 33, 34]. *Dst-e* exon 23 FSD+*jeb* mutants by contrast displayed a sharp disparity: both tail phenotypes exhibited amelioration of disease (higher tension and later lesion

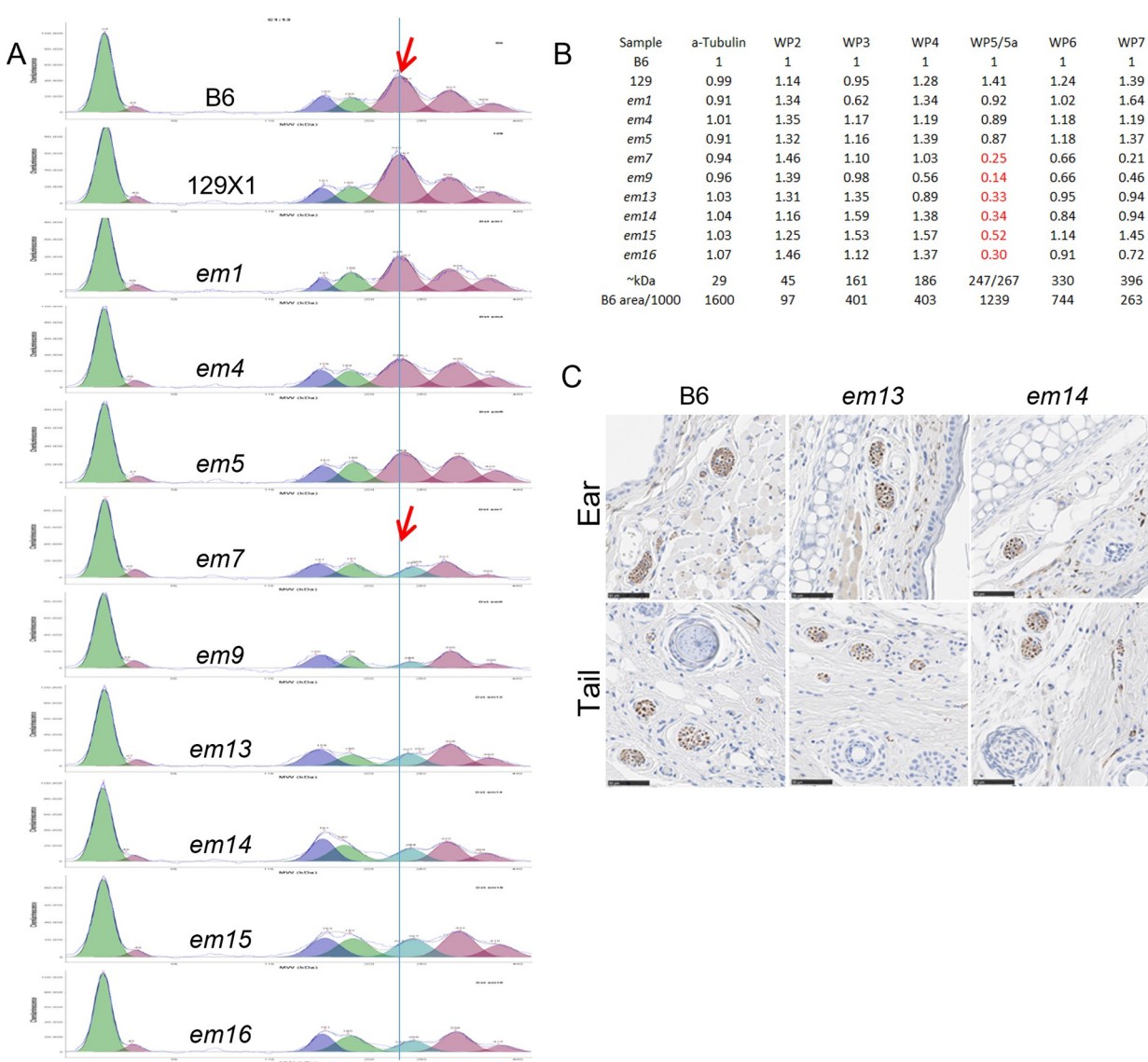

| Sample | a-Tubulin | WP2 | WP3 | WP4 | WP5/5a | WP6 | WP7 |
|---|---|---|---|---|---|---|---|
| B6 | 1 | 1 | 1 | 1 | 1 | 1 | 1 |
| 129 | 0.99 | 1.14 | 0.95 | 1.28 | 1.41 | 1.24 | 1.39 |
| em1 | 0.91 | 1.34 | 0.62 | 1.34 | 0.92 | 1.02 | 1.64 |
| em4 | 1.01 | 1.35 | 1.17 | 1.19 | 0.89 | 1.18 | 1.19 |
| em5 | 0.91 | 1.32 | 1.16 | 1.39 | 0.87 | 1.18 | 1.37 |
| em7 | 0.94 | 1.46 | 1.10 | 1.03 | 0.25 | 0.66 | 0.21 |
| em9 | 0.96 | 1.39 | 0.98 | 0.56 | 0.14 | 0.66 | 0.46 |
| em13 | 1.03 | 1.31 | 1.35 | 0.89 | 0.33 | 0.95 | 0.94 |
| em14 | 1.04 | 1.16 | 1.59 | 1.38 | 0.34 | 0.84 | 0.94 |
| em15 | 1.03 | 1.25 | 1.53 | 1.57 | 0.52 | 1.14 | 1.45 |
| em16 | 1.07 | 1.46 | 1.12 | 1.37 | 0.30 | 0.91 | 0.72 |
| ~kDa | 29 | 45 | 161 | 186 | 247/267 | 330 | 396 |
| B6 area/1000 | 1600 | 97 | 401 | 403 | 1239 | 744 | 263 |

**Fig 5.** (A) capillary immunoelectrophoresis using Bioworld polyclonal anti-dystonin antibody and a-tubulin control (left peak). (B) areas of each peak in (A) normalized to B6 and to a-tubulin ((sample area/B6 area)/(sample a-tubulin area/B6 a-tubulin area)). Fractions in red are 5a 267 kDa peak compared to B6 247 kDa peak which hides 267 kDa peak and is missing in FSD samples. From 3–4 day old female trunk skin. (C) UCHL1/ PGP9.5 labeling of peripheral nerve cell bundles in *em13* FSD, *em14* FSD and age and sex matched B6 controls fixed in 10% NBF (all males at 6 weeks old). All *Lamc2^{wt/wt}*.

onset and progression than controls) while ear scores indicated disease exacerbation in the same mice (earlier lesion onset and progression to a 'very affected' state, **Fig 3C and 3G–3K**). Trunk hair loss and lesions occurred earlier in FSD+*jeb* mice than in FSD mice (**Fig 9A**), indicating *Lamc2^{jeb/jeb}* accelerated the FSD trunk hair loss and lesion phenotype just as FSD accelerated the *Lamc2^{jeb/jeb}* ear lesion phenotype. Ear and tail scoring results indicate that differences in quantity or structural changes in dystonin protein have different consequences in these two different sites of (mostly) skin. Notably, FSD+*jeb* tail tension increases correlate with the number of exon 23 amino acids coded before a stop codon and inversely with the size of the deletion, as the two happen to coincide almost completely (**Fig 3C and 3D**). This is unexpected since all FSD+*jeb* exon 23 deletions are theorized to have the same result–

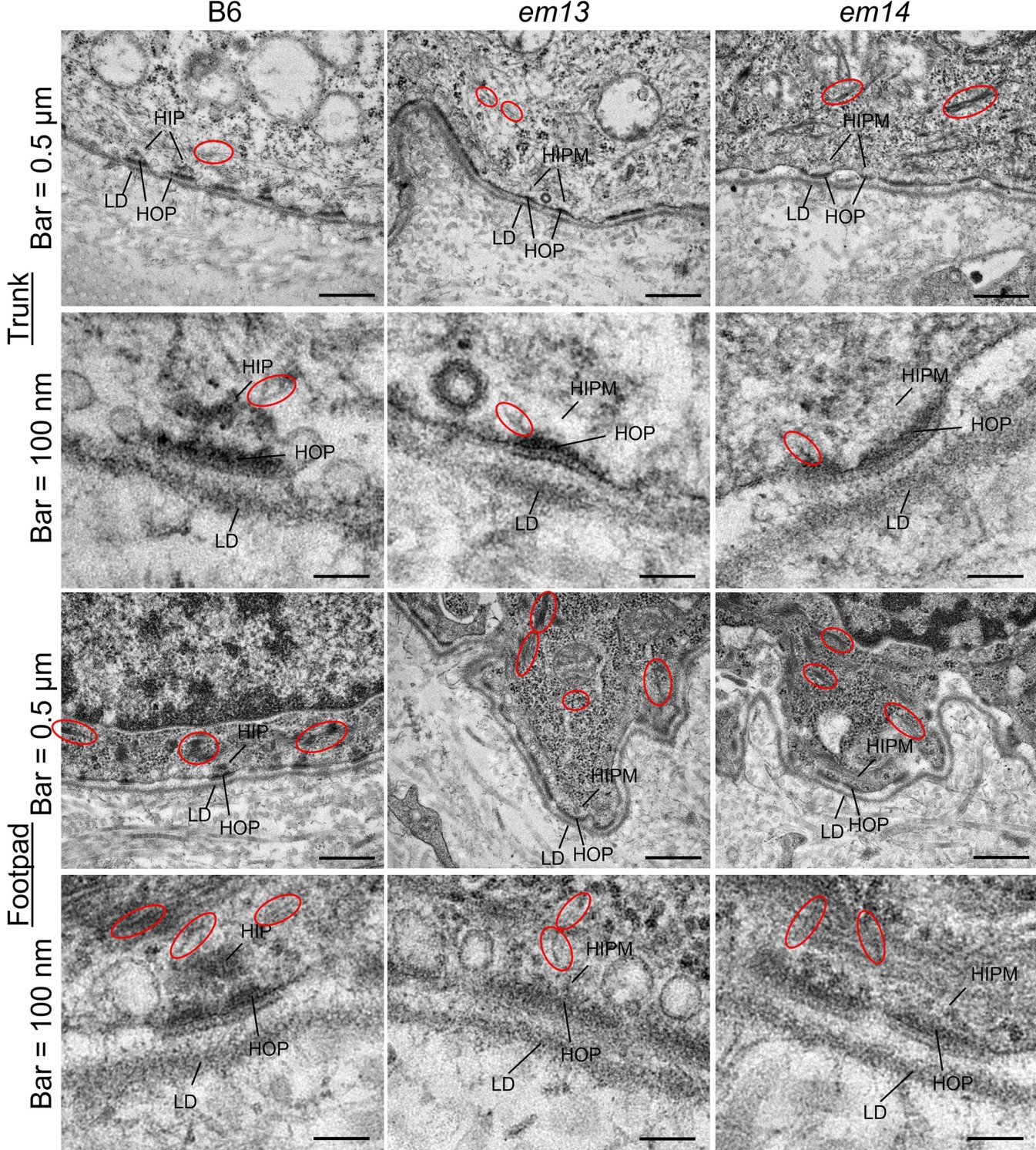

**Fig 6. Dorsal trunk and footpad TEM of 6 week old male B6, *em13* FSD and *em14* FSD at two magnifications.** Labeled elements: LD = lamina densa, HOP = hemidesmosome outer plaque, HIP = hemidesmosome inner plaque, HIPM = hemidesmosome inner plaque missing, red circles are intermediate filaments. All *Lamc2^wt/wt^*.

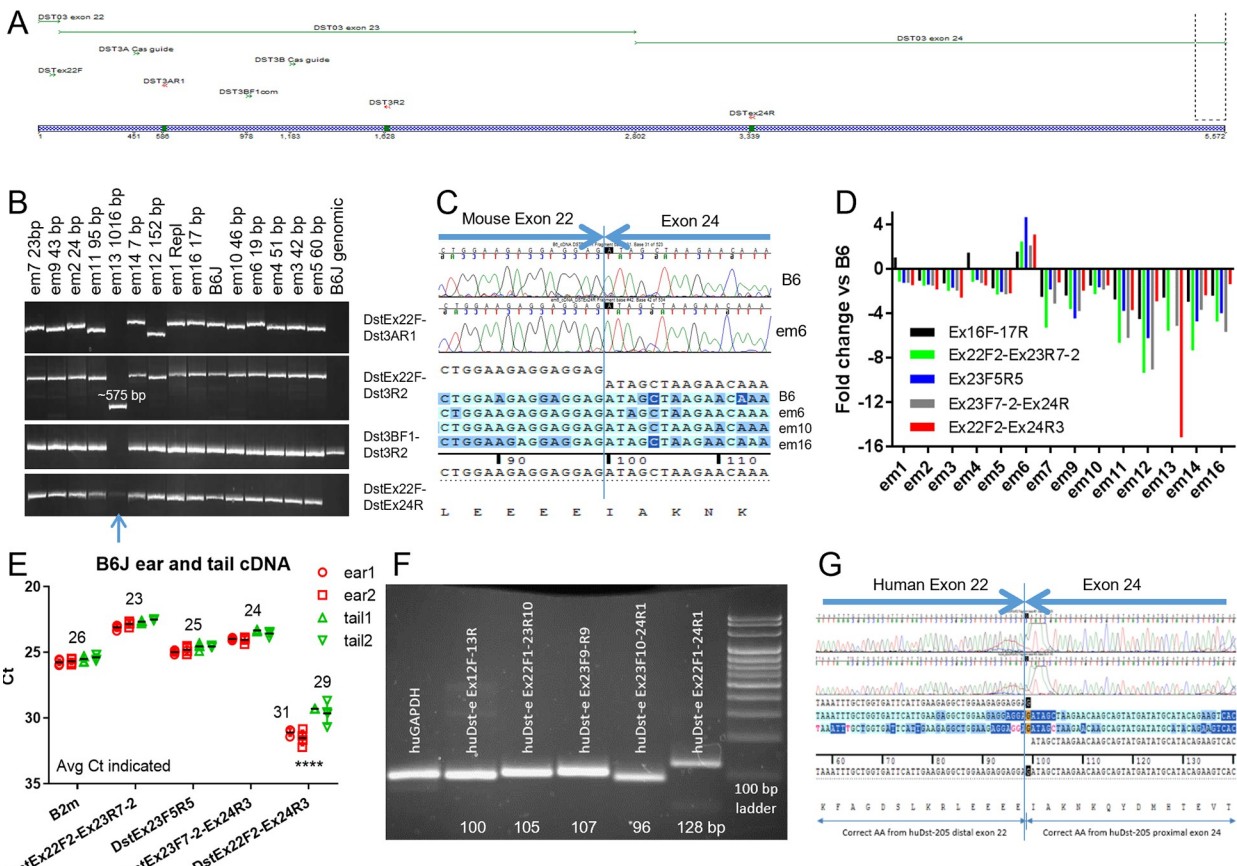

**Fig 7. Addressing DST-ES.** (A-B) cDNA from most *Dst em* lines were PCR tested with primers spanning mouse exon 22 to 23 (DstEx22F-Dst3AR1 and DstEx22F-Dst3R2), starting and end in exon 23 (Dst3BF1-Dst3R2) and spanning exon 22 to 24 (DstEx22F-DstEx24R), based on mouse Dst-213 exons. B6 wild-type cDNA product sizes are 544, 1591, 672 and 597 bp respectively, where 597 bp 22F-24R size is if 2703 bp in-frame exon 23 is skipped (**Table 1**). Blue arrow indicates weak but present ex22F-24R band in *em13*. (C) Sequence of mouse cDNA Ex22F-24R products completely skip exon 23, including for B6 wild-type. (D) Fold change compared to B6 for tail skin cDNA from various *Dst em* lines (n = 1 sample per line, 4 technical replicates each). (E) cDNA from equal quantities of RNA extracted from 11 week old B6 male ears and tails show ~5 cycle delay/32 fold less Ex22F-24R product than exon 23 containing RNA product in tail and ~7 cycle delay/128 fold less in ear. Ear and tail statistically differ (**** is p<0.0001) for Ex22F-24R (ear ~2 cycle/4 fold less) but not for others, ruling out differential loading (1 sample per tissue per mouse, 4 technical replicates each). Error bars are SD. (F) PCR of human adult skin cDNA using human *DST* specific primers also reveals presence of an exon 23 skipping product in human *DST-e*/GRCh38 DST-205 (ENST00000370765.11, 2649AA), confirmed by sequencing in (G).

elimination of the whole DST-e protein or at least skipping of the exon–so deletion size within the exon should not matter.

Histological evaluations of FSD+*jeb* at 16–18 weeks of age were compared to B6, FSD and B6-*Lamc2^{jeb/jeb}* controls. In both ears and tails, there was less dermal-epidermal separation in FSD+*jeb* than in B6-*Lamc2^{jeb/jeb}* controls, suggesting stronger junctional adhesion (**Fig 9G and 9H**). However, this observation did not align with 'scores' measurements of disease severity in either tissue. For ears, scores reveal that FSD+*jeb* was worse than B6-*Lamc2^{jeb/jeb}* (**Fig 3G**), but histological evaluation of percent of dermal-epidermal separation suggested the opposite (**Fig 9H**). For tails, scores for FSD+*jeb* were exceptionally protective (onset at 30+ wks vs 10 wks for B6-*Lamc2^{jeb/jeb}*) while histomorphometry indicated that FSD+*jeb* was protective, but not to the same degree (**Fig 9B, 9C, 9G and 9H**). Thus, percent of dermal-epidermal separation indicated stronger attachment in both ears and tails, but that attachment strength alone did not correlate with disease severity or progression.

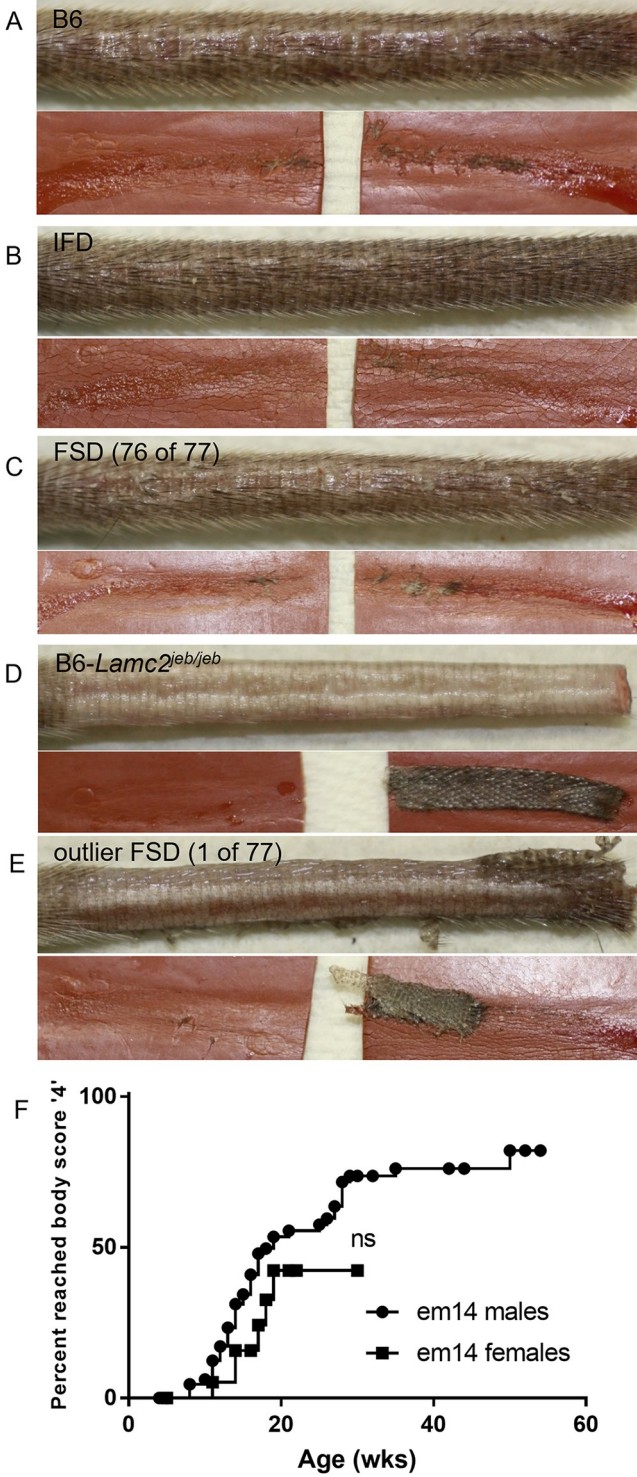

**Fig 8. Typical tail tension test skin removal.** (A) B6, (B) IFD, (C) FSD (same images as 2G-H), (D) B6-*Lamc2*$^{jeb/jeb}$. (E) atypical large amount of skin removed from FSD (n = 1 of 77) never seen in B6 (n = 0 of 55) or IFD (n = 0 of 18). All mice are males. All mice imaged are *Lamc2*$^{wt/wt}$ except (D) *Lamc2*$^{jeb/jeb}$. All show tension test results at 20 weeks except (D) at 10 weeks. Each image pair shows the tail and the rubber pads used to clamp the tail during the tension test, as they appear after the test is performed. (F) FSD *em14* (*Lamc2*$^{wt/wt}$) age when body score first reaches '4' (moderately affected) shows a trend for males to be affected earlier than females which does not reach statistical significance.

**Table 1. Primers.**

**Regular PCR for mouse Dst em genotyping**

| Dst primers used | DNA size (bp) | used for mutation # | primer name | sequence |
|---|---|---|---|---|
| ReplF-R1 | 152 | em1 | Dst3AReplF | GAGGCGGAAGCACAAAGG |
| wtF-R1 | 152 | | Dst3AwtF | GCGGAAGCTCGGCGA |
| F4-R4 | 169 | em2-10 | Dst3F2 | ACATTCGCTGGCAACCGA |
| F2-R1 | 704 | em11-12 | Dst3AF4 | AGTACCGCCGTGAGCTGGA |
| F2-R2 | 1751 | em13 | Dst3BF5 | CCTATCAGTTGGAATTAGAGTCTCTCC |
| F5-R5 | 96 | em14-16 | | |
| | | | Dst3AR1 | ACCAGGGCCCGTTTCTG |
| | | | Dst3R2 | AAGTGTTCGTTGGTGTTCCGTA |
| | | | Dst3AR4 | GTCTGCCTGGTGAAGGTGTTT |
| | | | Dst3BR5 | TCCTCTCGGCTAGTGCGTG |

**For mouse cDNA PCR and sequencing**

| | cDNA size (bp) | | | |
|---|---|---|---|---|
| Ex22F-Dst3AR1 | 544 | | DstEx22F | AATTTGCTGGTGATTCGTTGAA |
| Ex22F-Dst3R2 | 1591 | | Dst3BF1 | GGACGAGACGAATAACACACTCAA |
| Dst3BF1-Dst3R2 | 672 | | Dst3AR1 | ACCAGGGCCCGTTTCTG |
| Ex22F-24R | 597** | | Dst3R2 | AAGTGTTCGTTGGTGTTCCGTA |
| | | | DstEx24R | CAGGATCAACGATGCCATGA |

**if skips exon 23

Where used, mouse exon numbers are based on Dst-e/Dst-213 in Ensembl.

**For mouse cDNA qPCR**

| Ex16F-17R | 142 | | Ex16F | CAGATCTACTCTATGTCTTCCACGTACA |
|---|---|---|---|---|
| Ex22F2-Ex23R7-2 | 105 | | Ex17R | TCTTGTCTGCTATAACTGCTTCCTCTT |
| Ex22F2-Ex24R3 | 160 | | Ex22F2 | CGCAGTATATTAAATTTGCTGGTGA |
| Ex23F5R5 | 96 | | Ex23F7-2 | GGAGTTGGTGAAGCTTTTGACA |
| Ex23F7-2-Ex24R3 | 139 | | Ex23R7-2 | GCAGTAGATCTGAATACGCCCC |
| 18s | 178 | | Ex23F5 | CCTATCAGTTGGAATTAGAGTCTCTCC |
| B2m | 104 | | Ex23R5 | TCCTCTCGGCTAGTGCGTG |
| | | | Ex24R3 | GTTCTGCACCCTTCCAGCAT |
| | | | 18s-F | CCGCAGCTAGGAATAATGGAAT |
| | | | 18s-R | CGAACCTCCGACTTTCGTTCT |
| | | | B2mF | TTCTGGTGCTTGTCTCACTGA |
| | | | B2mR | CAGTATGTTCGGCTTCCCATTC |

**For human cDNA PCR and sequencing**

| Ex12F-13R | 100 | | Ex12F | CCATCTGTGTGCTTCACCGT |
|---|---|---|---|---|
| Ex22F1-23R10 | 105 | | Ex13R | CATGCCAAAGAGTCAGGACATTC |
| Ex23F9R9 | 107 | | Ex22F1 | ACTGCCCTGGTCACTCTCATG |
| Ex23F10-24R1 | 96 | | Ex23F9 | CACAAGGGCACATGCTGTAGC |
| Ex22F-24R1 | 128** | | Ex23R9 | TTCTCTGGCAGATTTGTAGTCTTTCTAA |
| huGAPDH-F | 101 | | Ex23F10 | CGCATGTGAAATGGAACTGGT |
| | | | Ex23R10 | ATGTTCAGAAGTCTCCTTACACCTTTT |
| | | | Ex24R1 | TGTTTTAATGTTGTGACTTCTGTATGCA |
| | | | huGAPDH-F | ACAACTTTGGTATCGTGGAAGG |
| | | | huGAPDH-R | GCCATCACGCCACAGTTTC |

**if skips exon 23

Where used, human exon number based on DST-e/DST-205

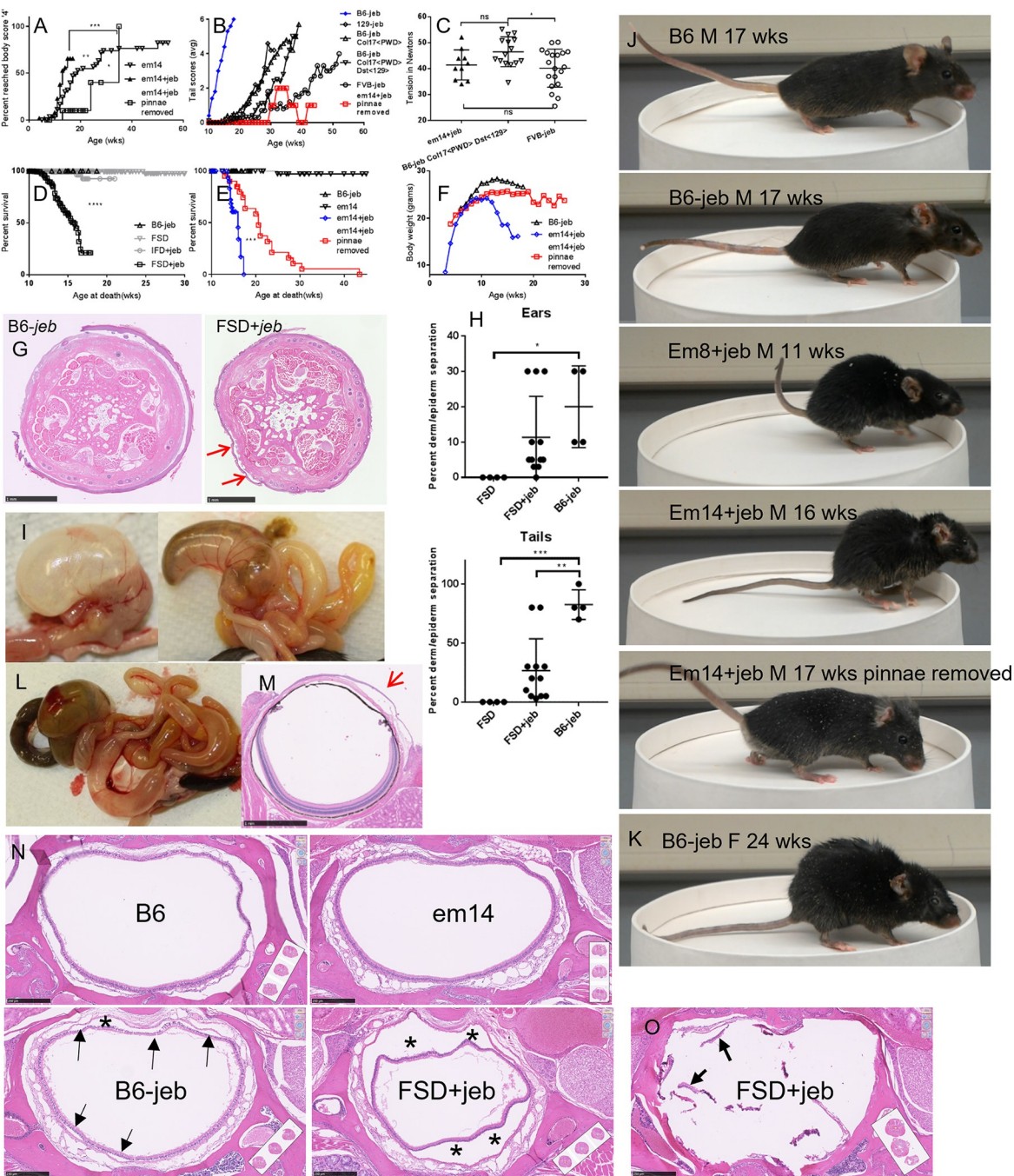

**Fig 9. FSD+*jeb* notable disease acceleration, pinnae removal correction and addition phenotypes beyond tension and scores.** (A) *em14* FSD+*jeb* have accelerated body symptom (hair loss and lesions) compared to *em14* FSD alone (p<0.01). Symptoms are greatly ameliorated when FSD+*jeb* pinnae are removed (p<0.05 vs *em14*, p<0.001 vs *em14+jeb*). (B) *em14* FSD+*jeb* with pinnae removed tail scores compared to controls. (C) 10 week male tail tension comparison of the three *Lamc2*^*jeb/jeb*^ strains with latest tail onset from (B): *em14* FSD (pinnae removed), B6-*Col17a1*^*PWD/PWD*^ *Dst*^*129/129*^ double congenic and FVB. Error bars are SD. (D) Early mortality of FSD+*jeb* compared to FSD or B6-*jeb* alone (data combines both sexes and lines *em8* and *em14*. (E) *em14* FSD+*jeb* survival with and without pinnae removal, p<0.0001. (F) Body weight comparisons show pinnae removal from *em14* FSD+*jeb* improves weight. (G) H&E sections of 16–18 week old male B6-*jeb* tails typically exhibit >50% dermal/epidermal separation but age and sex matched FSD+*jeb* exhibit much less separation. (H) Graphical representation of amount of dermal-epidermal separation in H&E sections in ears and tails from 16–18 week old males shown in (g), combining lines *em8* and *em14*. Error bars are SD. (I) Absence of food and presence of gas in digestive tract as mice sicken toward death. (J) Photo demonstration of typical health deterioration in FSD+*jeb* compared to B6 control and improvement when pinnae removed. (K-L) Occasional B6-*jeb* females at 24 weeks have external symptoms (K) and digestive tract bloating (L) like FSD+*jeb* at 16 weeks. (M) Anterior corneal epithelium separations at the lamina limitans anterior (Bowman's layer) from the substantia

propria are common in FSD+*jeb*. (N) Typical examples of mid-nasopharynx comparing B6 at 19 wks, *em14* FSD at 24 wks, B6-*jeb* at 14 wks and *em14* FSD+*jeb* at 16 wks. Arrows and * point to epithelial separation from connective tissue. Other voids shown are in other layers. (O) FSD+*jeb* occasionally exhibit extreme nasopharynx disintegration in H&E cross-sections (example shown at 14 wks old). Arrows point to residual epithelial tissue. B6-*jeb* is B6-*Lamc2*^*jeb/jeb*.

Comparison of *Dst* FSD heterozygotes (*em/wt*) to *em/em* and *wt/wt*, each in combination with *Lamc2*^*jeb/jeb*, for 4 different FSD lines (*em8*, *em10*, *em13* and *em14*) revealed the following phenotype trends: For overall body condition, *em/wt* matched *wt/wt* (*wt* dominant); for tail scores, *em/wt* was intermediate (haploinsufficient); and for ear scores *em/wt* gave improved results compared to both *em/em* and *wt/wt*. Tension tests gave different results for three different lines. This might be due to 'noise' or might represent genuine differences based on deletion size and location (**Fig 10**). Similar testing of a single IFD line (*em2*) in a heterozygous state also

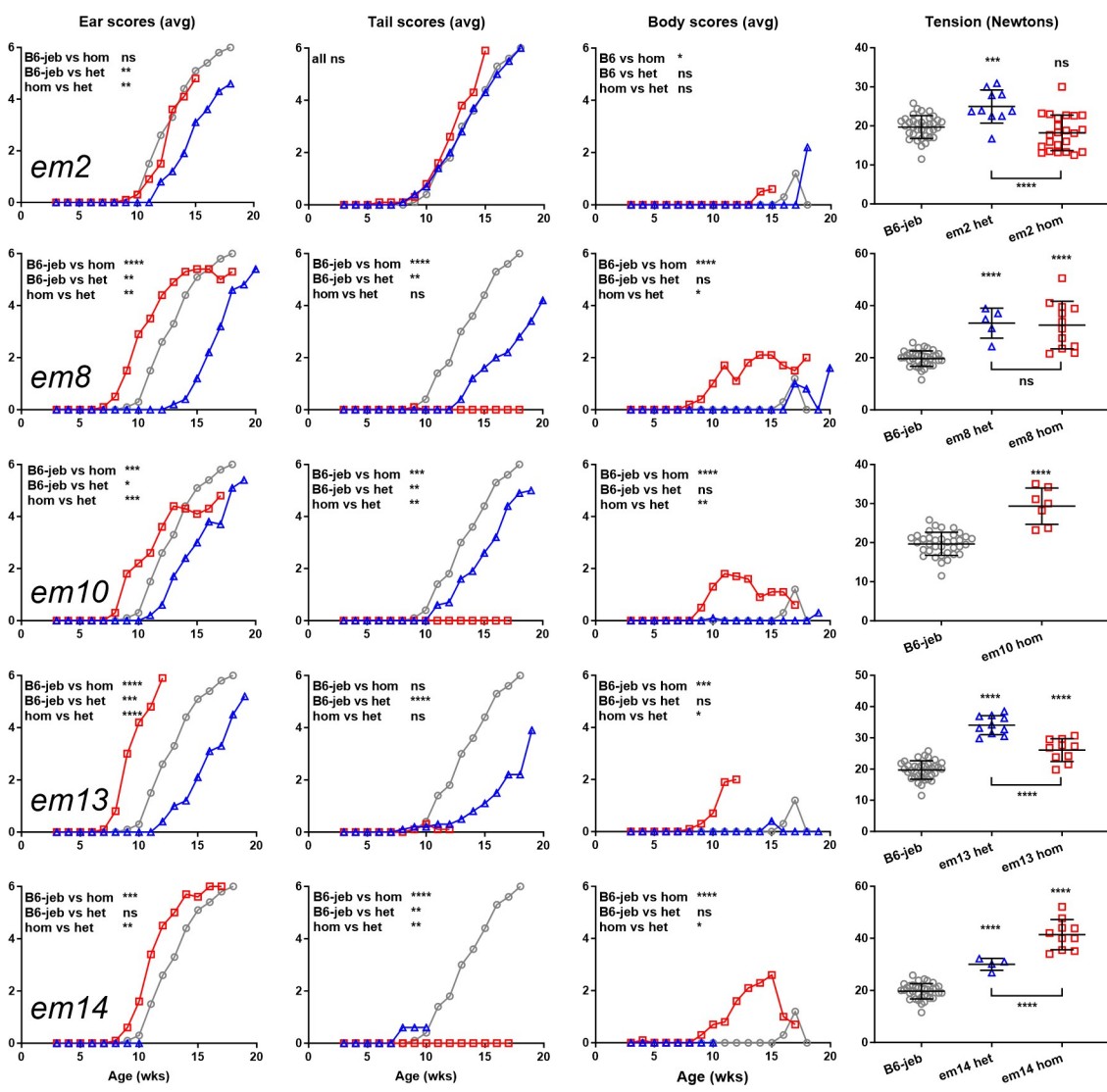

**Fig 10. *Dst*^*em* het vs hom and wt results when *Lamc2*^*jeb/jeb* for select lines.** Marker symbols are the same across each line. B6-*Lamc2*^*jeb/jeb* (B6-jeb) controls are the same for each *em* line. B6-jeb open circle, het open triangle, hom open square for all. *Em2* is IFD, *em8, 10, 13* and *14* all FSD. All points are male *Lamc2*^*jeb/jeb*. Tension is at 10 weeks of age. Tension stats shown above values are vs B6-jeb, below are het vs hom. Tension error bars are SD. Ear, tail and body scores stats are calculated from survival age each mouse reaches a score of 4 ('moderately affected').

showed an apparent allelic synergy that improved results for ear scores and a suggestive but not statistically significant trend toward allelic synergy for tension while tail and body scores did not distinguish between any alleles.

Aside from the ear, tail and trunk phenotypes typically noted and studied in various *Lamc2-jeb/jeb*, *Col17a1emDcr* and other EB mouse models, FSD+*jeb* adult mice displayed striking and unexpected overt disease symptoms including weight loss, hunched posture and labored breathing starting ~10 weeks of age requiring that all be euthanized as moribund by 18 weeks of age (**Fig 9D and 9J**). Necropsy of hunched FSD+*jeb* mice consistently revealed gaseous bloating of the stomach, small intestines and/or large intestines (flatus, **Fig 9I**) but no other gross abnormalities. Diarrhea was not observed. These are consistent with mice not eating probably due to extensive blistering lesions affecting multiple systems. These 'wasting' symptoms were not evident in parental lines FSD to one year of age or previously noted in B6-*Lamc2jeb/jeb* mice prior to required euthanasia for lesions at ~20 weeks of age (**Fig 9J**). To investigate further, female B6-*Lamc2jeb/jeb* mice, which develop ear and tail lesions later than their male counterparts and therefore could be kept alive longer [33], were aged. Two of six (~33%) developed similar symptoms to FSD+*jeb* (hunched, moribund, flatus) at ~22–27 weeks of age (**Fig 9K and 9L**), suggesting that FSD+*jeb* wasting could be an acceleration of a phenotype found in B6-*Lamc2jeb/jeb* mice that was not previously documented.

In an attempt to identify and characterize the source of the wasting symptoms accelerated in FSD+*jeb*, several experiments were performed trying to investigate bacterial infection or internal dermal-epidermal separation causes: FSD+*jeb* mice were treated with either Sulfamethoxazole-Trimethoprim or ampicillin in their drinking water. Both were without effect. Histological sections of lungs from affected mice were stained with Gomori's Methenamine Silver (GMS) and tissue gram stains, but no bacterial or fungal growth was found. Ears and tails were cultured for bacteria but no pathological organisms were identified. Histological examination of the digestive tract (stomach, small intestine, large intestine) found no abnormalities associated with the gross finding of flatus (gaseous bloating) nor any evidence of *Clostridia spp.* infections. Histological examination of dorsal full-body cross-sections from the muzzle to top of the rib cage was done primarily to look for defects in the mucus membranes to explain dyspnea (difficulty breathing) and anorexia (not eating). The most obvious symptoms were epithelial lining separation along the nasopharynx of FSD+*jeb* mice. B6-*Lamc2jeb/jeb* exhibited frequent but much less extensive separation while FSD alone rarely exhibited any separation and B6 controls never did (**Fig 9N**). While H&E of FSD+*jeb* nasopharynx cross-section usually demonstrated a normally shaped intact airway no matter how much separation from the underlying connective tissue, occasional complete breakdown of the airway lining was observed in some specimens (**Fig 9O**). FSD+*jeb* exhibited lesser amount of separation of olfactory epithelium and occasional separation in the oral cavity. Lamina propria-epithelial separation was never noted in the esophagus or trachea (**Fig 11**). Larger lamina propria-epithelial separations were also common in cornea, a region of high *Dst-e* expression (**Fig 9M**). No visible lesions were found in other organs.

**Pinnae removal ameliorates FSD+*jeb* trunk hair loss/blistering and wasting/moribund phenotypes.** FSD+*jeb* mice which did not get overly sick and moribund still required euthanasia by 18 weeks of age due to the severity of ear lesions. At that point tail lesions were still not visible, a feature considered to be very unusual for B6-*Lamc2jeb/jeb* mice and an indicator of a very strong *Dst* FSD 'protective' role in the tail. In an attempt to extend usable lifespan to the point that age of tail lesion onset and progression could be determined, female and male *em14* FSD+*jeb* had their pinnae surgically removed at 4–7 weeks of age. Removal of the pinnae virtually eliminated erosions associated with that region of the skin (**Fig 9J**). These same mice had unexpected significantly delayed development of trunk hair loss/blistering compared to

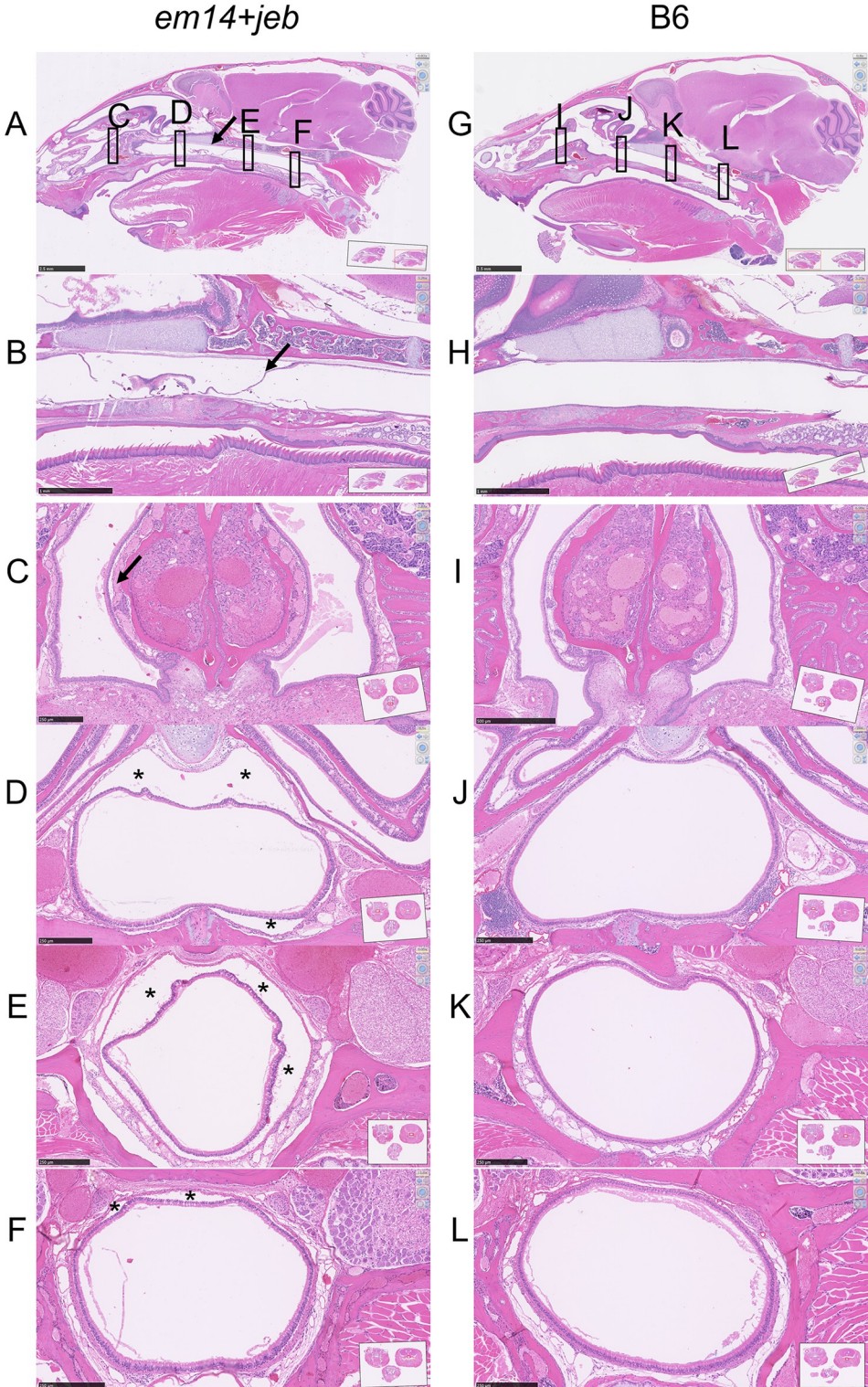

**Fig 11. Nasopharynx.** *Em14+jeb* mouse sagittal head view at 16 weeks of age (A) zoomed in on proximal nasopharynx in (B). Cross-sections from a separate *em14+jeb* mouse in (C-F) with approximate locations indicated in (A). (G-L) are comparable sections from B6 control at 14 weeks of age. Arrows and * indicate areas of lamina propria/respiratory epithelial separation.

FSD+*jeb* and FSD alone (**Fig 9A**)—perhaps due to decreased scratching of the areas with the primary pruritic areas removed—and wasting/moribund symptoms compared to FSD+*jeb* untreated mice (pinnae intact) (**Fig 9E, 9F and 9J**). While life span is significantly extended, 95% of mice still required euthanasia by 30 weeks of age due to wasting symptoms (**Fig 9E**), so tail score data beyond that point were not robust. To the extent that data are available, late tail scores from *em14* FSD+*jeb* mice suggested more 'protection' against disease than any other *Lamc2^{jeb/jeb}* genetic variable identified to date, including highly resistant FVB-*Lamc2^{jeb/jeb}* mice (**Fig 9B**). Because pinnae removal successfully prevented ear erosions and because FSD +*jeb* are genetically protected from developing tail erosions until quite late in life, FSD+*jeb* with pinnae removed has inadvertently proven to be a model focused on visceral symptoms of EB without the confounding or limiting influence of severe skin disease.

## Discussion

In contrast to full protein loss-of-function (null) mutations, such as a neonatal lethal mutation *Lamc2^{tm1Uit}* [33, 35], genetic backgrounds substantially influences the severity of the hypomorphic mouse *Lamc2^{jeb/jeb}* disease. This modifier effect is attributed to at least 7 QTL, one of which has been formally identified as *Col17a1* [13, 34]. An initial goal here was to use CRISPR/Cas9 replacement to address the hypothesis that a phenotypic modifier of *Lamc2^{jeb/jeb}* induced JEB disease QTL mapping to mouse chr1 near *Dst-e* (chr1:24-38Mb, peak 34Mb) was caused by innocuous amino acid changes in that gene that distinguish mouse strains B6 and 129. Successful conversion of the amino acid at p1226 in *Dst-e* (Ensembl Dst-213) from the B6 to the 129 allele (R→Q) resulted in *Lamc2^{jeb/jeb}* phenotypes midway between the two parental strains, confirming its involvement but indicating that other polymorphisms also play a role (**Figs 1 and 2**). As the attempted *Dst-e* p1469 G/V replacement was not successful, it remains possible that this polymorphism might account for the remainder of this B6/129 phenotypic difference.

We also took advantage of founder mice arising from this gene targeting experiment that carried deletions in *Dst-e*. The DST-e protein includes three domains of comparable size: a plakin domain, a central coiled-coil rod and an intermediate filament binding domain (IFBD) [18]. P1226 and p1469 both map to *Dst-e* exon 23, which exclusively produces the coiled-coil rod, and the deletions identified all map entirely within that large exon. Four nested in-frame deletions (IFD) and 11 frame-shift deletions (FSD) split between the two CRISPR/Cas9 cut sites were propagated (**Fig 4**). Each was made homozygous on a B6 background (B6-*Dst-e^{em/em}*) to test for stand-alone phenotypes and on a B6-*Lamc2^{jeb/jeb}* background (B6-*Lamc2^{jeb/jeb}* *Dst-e^{em/em}*) to test for modifier effects.

B6-*Dst-e^{em/em}* IFD mice were indistinguishable from parental strain wild-type B6 by all methods tested. B6-*Dst-e^{em/em}* FSD mice developed ear lesions and blisters on the dorsal skin consistent with EB simplex. No mice exhibited signs of dystonia musculorum (DM), which is also caused by mutations in *Dst*. This is expected since the exon mutated in this study (*Dst-e*/Dst-213 exon 23) is not encoded by the nerve or muscle isoforms *Dst-a* and *Dst-b*. The lack of DM means these mice live much longer and avoid confounding phenotypes when used in studies focused on EB simplex.

IFD are anticipated to produce DST-e protein at normal levels but with slightly shortened coiled-coil segments (native ~901AA reduced by 20AA in largest 60bp IFD). FSD are expected to result in premature stop codons (PSC) disrupting DST-e protein production. PCR, qPCR and sequencing of cDNA indicate *Dst-e* RNA is made at a normal level in FSD, IFD and controls, but capillary immunoelectrophoresis shows that the most abundant 247 kDa protein product is not present in FSD mice (**Figs 5 and 7**). The 247 kDa isoform should be DST-e, the

only protein documented to utilize the Dst-213 exon 23 which is disrupted by the mutations described here (ensembl.org).

PCR, qPCR and sequencing of cDNA using Dst-213 exon 22 forward and exon 24 reverse primers to determine if FSD stop codons in in-frame exon 23 (2703 bp, 901AA) led to exon skipping and a shorter rod-less DST-e related protein. Unexpectedly based on Ensembl and the majority of literature, a cDNA product coding from *Dst-e* exon 22 to 24 (skipping exon 23) was found not only in FSD, but also in IFD and B6 controls (**Fig 7**). This product is first mentioned in Okumura 2002 but rarely since and is not recognized in Ensembl (ensembl.org) [32]. Plectin and desmoplakin, which are both structurally and functionally similar to epithelial dystonin, produce both rod *and* rodless versions of the protein, supporting the possible existence of a rodless version of DST-e [36–38]. Because of unusual FSD modifier phenotypes discussed below, rodless DST-e was considered further. Based on nomenclature conversion in recent years from BPAG1e to DST-e and Okumura 2002 nomenclature BPAG1eS, it is here called DST-eS. Similar testing of human adult skin cDNA with Ensembl human Dst-205 exon 22, 23 and 24 primers confirmed the existence of an exon 23 skipping *DST-eS* RNA product in humans as well (**Fig 7**).

*Dst-eS* RNA is less abundant than *Dst-e* in skin, with a significant difference between ear and tail skin (~32 fold less than *Dst-e* in tail and ~128 fold less in ear). In addition to the 247 kDa protein eliminated in FSD, capillary immunoelectrophoresis identified four other peaks in the range 2–440 kDa. Either of the peaks noted at 161 and 186 kDa could be DST-eS. Area-under-the-curve indicates relative abundance of both 161 and 186 kDa products in trunk skin to be about one-third of the 247 kDa putative DST-e protein in B6, 129X1, the *em1* replacement and *em4* IFD. Similar quantities of the 161 and 186 kDa proteins are present in FSD lines, in which the presumed DST-e 247 kDa product is absent. All other protein peaks besides the 247 kDa peak are not notably altered in size or position in IFD or FSD. Removal of the 247 kDa peak in FSD mice reveals an underlying less abundant 267 kDa product not visible but likely present in samples from IFD and B6 control mice (**Fig 5**). It is expected that only dystonin isoforms which include Dst-213 exon 23 will be affected by the mutations described here. Per Ensembl, no mouse isoform besides *Dst-e*/Dst-213 is known to include this exon. DST-eS to DST-e ratios may differ between ear skin and tail skin, reflecting cDNA qPCR differences (*Dst-eS* RNA 4-fold more abundant in tail skin than in ear skin) and explaining phenotypic differences, but protein from these tissues was not evaluated.

In previous studies of the *Lamc2*$^{jeb/jeb}$ hypomorphic mouse model, phenotypic testing by ear scores, tail scores and tail tension tests have always shown concordant patterns [14]. *Dst-e* AA replacement and IFD models modify the *Lamc2*$^{jeb/jeb}$ disease in agreement with these patterns while FSD exhibit previously unseen contradictory phenotypes, indicating a more complicated digenic interaction. One possibility is that DST-e, which is removed in FSD mice, and DST-eS which is (proposed to exist and be) still present both play roles in hemidesmosomal dermal-epidermal adhesion but their roles are not identical.

When compared to B6-*Lamc2*$^{jeb/jeb}$ controls, B6-*Lamc2*$^{jeb/jeb}$ *Dst-e*$^{FSD/FSD}$ accelerate disease by most measures and in most tissues: ear scores, body condition scores, survival, body weight and histological evaluation of nasopharynx, digestive tract and cornea. But B6-*Lamc2*$^{jeb/jeb}$ *Dst-e*$^{FSD/FSD}$ conversely delays disease by tail measurements: tail condition score, tension test, and histology (**Figs 3, 8 and 9**). Further, when B6-*Lamc2*$^{jeb/jeb}$ *Dst*$^{FSD/wt}$ are included in the comparison, another unexpected result is observed. Body scores and tail scores display 'normal' but differing progressions–*Dst-e*$^{FSD/wt}$ body score is like *Dst-e*$^{wt/wt}$ (wt dominant) and *Dst-e*$^{FSD/wt}$ tail score intermediate between *Dst-e*$^{wt/wt}$ and *Dst-e*$^{FSD/FSD}$–but *Dst-e*$^{FSD/wt}$ ear scores are consistently better than both *Dst-e*$^{wt/wt}$ and *Dst-e*$^{FSD/FSD}$, indicating that ear skin

tissue (but not trunk or tail) benefits by a mix of *Dst-e*^wt and *Dst-e*^FSD functionality (**Fig 10**). Heterozygotes tested *em2* IFD ear scores showed similar results.

To explain the ear heterozygous effect as a modifier of *Lamc2*^jeb/jeb phenotype, we propose that DST-e protein levels are affected by allelism: *Dst-e*^wt appears to make both DST-e and DST-eS proteins in an approximately 3:1 ratio, as evidenced in trunk skin (may differ in ear and tail). *Dst-e*^FSD produces only DST-eS, in the same levels as before (not altered when DST-e is absent) (**Fig 12**). *Dst-e*^FSD/wt hets would make a normal amount of DST-eS but half the normal level of DST-e. Therefore, ratios in *Dst-e*^wt/wt are 3:1, *Dst-e*^FSD/FSD are 0:1 and *Dst-e*^FSD/wt heterozygotes are 1.5:1. Thus it appears that a higher fraction of DST-eS in ears without total loss of DST-e provides a more optimum binding and better disease amelioration. In tails, by contrast, replacement of all DST-e with DST-eS in dermal-epidermal connections provides the best outcome. In trunk skin, DST-e provides better connections than DST-eS, though partial loss of DST-e is tolerated without effect.

Identification of genetic modifiers of EB in mice are useful for defining not only causes of clinical variability in severity between patients but also developing models of subtypes of EB with particular symptoms. The most common and severe phenotype is skin fragility (blistering due to dermal-epidermal separation). Secondary lesions include mucous membrane defects [17], nail abnormalities, pyloric atresia, other gastrointestinal issues, corneal defects and muscle weakness [26, 28, 39–42]. Some modifier mutations in *Col17a1*, when combined with the *Lamc2*^jeb hypomorphic allele (i.e. B6- *Lamc2*^jeb/jeb *Col17a1*^em1/em1), resulted in mice with nail abnormalities. These bear similarities to nail disease in humans with *Col17a1* mutation induced JEB, but are not evident in either B6-*Col17a1*^em1/em1 or B6-*Lamc2*^jeb/jeb parental strains [34]. *Dst* FSD+*jeb* mutations likewise result in novel or vastly accelerated mucous membrane deterioration, wasting disease and corneal involvement not seen in parental strain FSD or B6-*Lamc2*^jeb/jeb which may be relevant to human symptoms [26]. These findings begin to define the specific genetic effects that account for clinical variability between individual patients.

EB simplex in humans most often results from dominant mutations in *KRT5* or *KRT14*, whose products compose the intermediate filaments which crisscross basal keratinocytes, providing strength and anchoring to desmosomes and hemidesmosomes [43]. EBS can also be caused by mutations, usually recessive, in *PLEC*, *DST*, *DSP*, *TGM5*, *JUP*, *PKP1*, *EXPH5* and *KLHL24*, some of which may impact dermal-epidermal strength via signaling, rather than being structural components themselves [39, 44]. Epidermal dystonin (DST-e) is a structural component of hemidesmosomes, and mutations which disrupt *Dst-e* coding are understood to cause EBS by disrupting KRT5/14 intermediate filament connections to hemidesmosomes [22].

Dystonin is associated with two very different diseases in both mice and humans: epidermolysis bullosa simplex when the epidermis specific isoform *Dst-e* is disrupted and dystonia musculorum (DM)/HSAN6 when the nerve specific isoform *Dst-a* is disrupted. Previous mouse models of dystonin mutations induced EBS *Dst*^tm1Efu [22] and *Dst*^dt-23Rbrc [19] (henceforth called EBS+DM) mutate *Dst-e* and *Dst-a* shared exons, and therefore demonstrate both diseases. Most human patients, by contrast, have only been documented with mutations affecting *DST-e* or *DST-a*, not both [24, 28–30]. One patient has been reported compound heterozygous for a missense mutation altering *DST-a* and *DST-b* but not *DST-e* (H269R) and a premature termination codon mutation altering both *DST-a* and *DST-e* (R1296X, *DST-a*/Dst-202 exon 29, *DST-e*/Dst-205 exon 19) resulting in a more complex phenotype with apparently standard dystonin EBS skin issues but additional eye and mucosal tissue involvement and a milder neurological phenotype than HSAN6. Light microscopy and TEM data on skin were not presented in this case, making comparison to our model difficult [26]. FSD mice presented

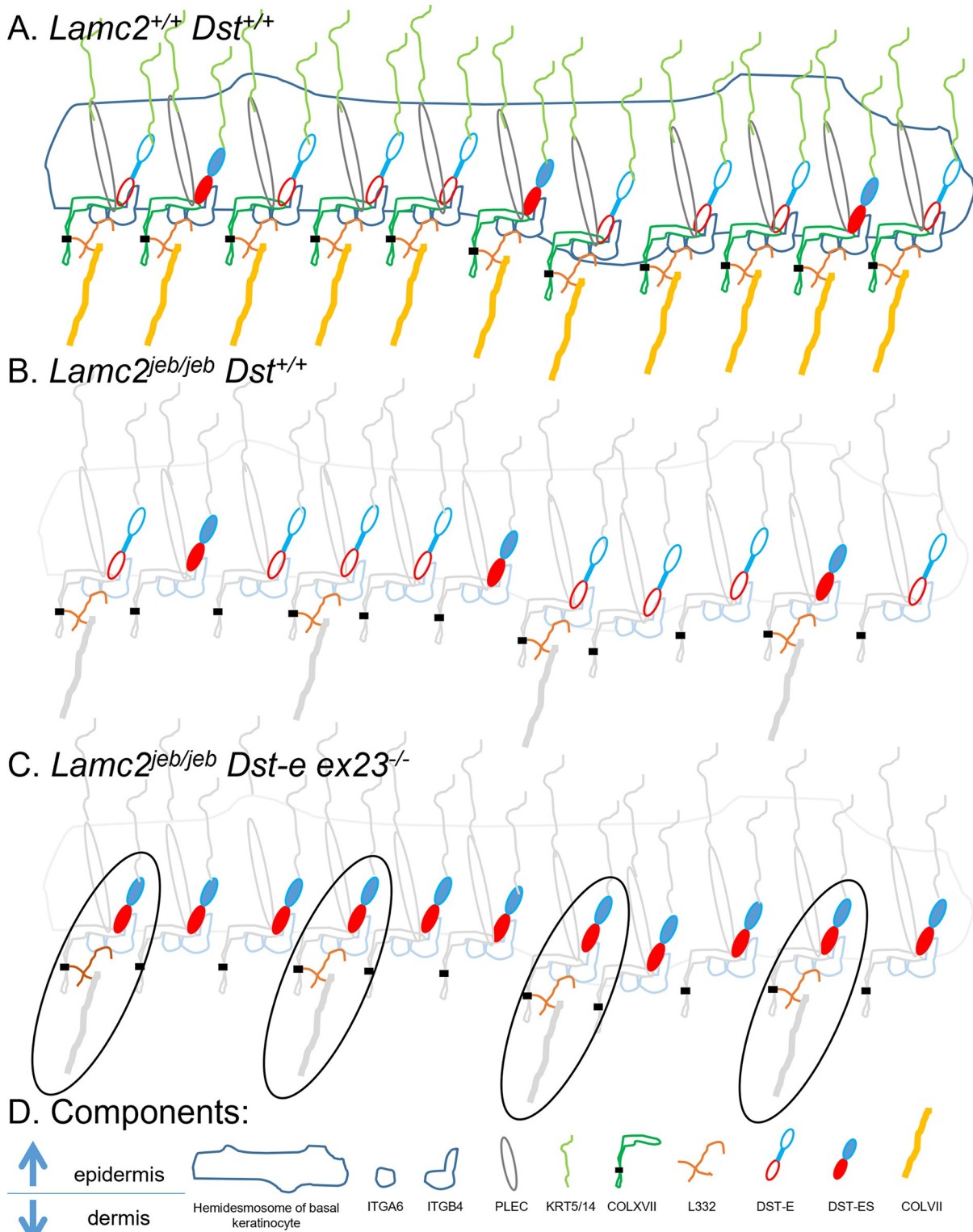

**Fig 12. Modeling of DST-e and DST-eS usage.** (A) If DST-eS is made at one-third the abundance of DST-e (based on capillary immunoelectrophoresis) then it will be incorporated in about one-fourth of normal hemidesmosomal connections. (B) Regular B6-*Lamc2^{jeb/jeb}*, showing that very little (~10%?) of the normal amount of healthy L332 is made, and does not occupy all otherwise normal hemidesmosomal binding sites, resulting in less binding strength. (C) If DST-e is absent but DST-eS is available at normal levels, it may be sufficient to provide a normal or close to normal number of binding sites available for L332 while L332 is available to bind in the same

frequency as in *Lamc2*<sup>jeb/jeb</sup>. 100% of L332 are bound to DST-eS complexes (circled), compared to about 3/4 to DST-e complexes and 1/4 to DST-eS in (B). (D) Orientation of epidermis and dermis at boundary and names of components shown in (A-C).

here are most akin to human dystonin induced EBS patients without nervous system disorders, which all have mutations in *DST-e* specific exon 23 (called hu-ex23), and contrast sharply with EBS+DM. FSD mice and hu-ex23 patients exhibit much milder EBS symptoms than those described for *Dst*<sup>tm1Efu</sup> and *Dst*<sup>dt-23Rbrc</sup> EBS+DM mouse models. EBS+DM mouse models exhibit skin fragility upon mechanical stress at a young age (2–4 weeks) not seen in FSD mice to 54 weeks old and seen to only a limited extent in hu-ex23 patients. EBS+DM early skin fragility is accompanied by dermal-epidermal separations visible by light microscopy. Dermal-epidermal separations have not been found in healthy skin from hu-ex23 patients and only found in FSD mice much later in life (>30 wks of age). Because of statements in publications of hu-ex23 mutations that 'keratin filaments extended to where the inner plaques should be, but did not seem to associate with any plasma membrane attachment structure' [28, 29], significant effort was made here to look for TEM evidence that the presence or absence of DST-eS when DST-e is deficient (i.e., defects in exon 23 vs *Dst-a*/*Dst-e* shared exons) might cause visible structural differences associated with a presence or absence of keratin IF attachment to hemidesmosomes. TEM samples from FSD homozygotes were compared to *Dst*<sup>dt-23Rbrc</sup> homozygotes by an author associated with both projects (Dr. Kabata). Both are missing the hemidesmosome inner plaque present in wild-type controls. Filaments which are probably keratin IF are visible extending to some HD outer plaques in some regions of *em13* and *em14* FSD samples but side-by-side comparison could not declare that these structures were missing from *Dst*<sup>dt-23Rbrc</sup> samples, or that any other clear differences existed between them. Despite a lack of concrete evidence, the contrast in phenotypes suggests that while all cause loss-of-function of *Dst-e*, the impact of EBS+DM models is more pronounced in skin. This can be explained by *Dst-eS*, which is identical to *Dst-e* except for skipping of exon 23, meaning that it would also be deleted in EBS+DM mice but not in FSD mice or hu-ex23 patients. Phenotype comparisons are summarized in **Table 2**.

Previous work suggests genetic modification of EB caused by mutations in *COL7A1*, *LAMC2* and *KRT14* are possible [5, 33, 45]. This study adds *DST-e* to the list, as *Dst-e* FSD EBS specific trunk symptoms were accelerated in the presence of *Lamc2*<sup>jeb/jeb</sup> (**Fig 9A**), though modification of *Dst-e* FSD EBS by innocuous *Lamc2* changes has yet to be demonstrated.

FSD+*jeb* exhibit a wasting disease including hunched posture and weight loss processing to a moribund condition and accompanied by flatus affecting the stomach, small intestine and/or large intestine by 14–18 weeks of age. Histological examination reveals substantial separation of epithelium from underlying connective tissue (lamina propria) throughout the nasopharynx with occasional lesser separation in olfactory and oral cavities (**Figs 9N, 9O and 11**). This likely accounts for the externally observed phenotypes, making breathing difficult and interfering with the desire to eat, leading to flatus, weight loss and eventual death. This appears to be an acceleration of a similar wasting phenotype only occasionally seen in B6-*Lamc2*<sup>jeb/jeb</sup> at >22 weeks old as severity of ear and tail lesions typically require euthanasia prior to that age. This is supported by the intermediate nasopharynx histological phenotype observed in B6-*Lamc2*<sup>jeb/jeb</sup> at ~17 wks of age. Thus, the B6-*Lamc2*<sup>jeb/jeb</sup> hypomorphic model includes a previously unreported mucous membrane/airway defect phenotype which is exacerbated more than the ear lesion phenotype when *Dst-e* is deficient (and much more than the tail blister phenotype, which is actually improved). Any future interest in studying this model of airway defects and accompanying wasting/moribund phenotype would be much more easily and reliably studied in FSD+*jeb* mice than in B6-*Lamc2*<sup>jeb/jeb</sup> particularly if the pinnae are surgically removed.

**Table 2. Phenotype comparisons.**

| organism | human | human | human | human | human | mouse | mouse | mouse | mouse | mouse |
|---|---|---|---|---|---|---|---|---|---|---|
| mutation | Q1124X | Q1124X | R1249X | c.15399delA PTC | H269R/ R1296X compound het | targeted deletion | Dst-a E1194X | 40kb deletion | 24–60 bp del | 7–1016 bp del |
| location | Dst-e exon 23 | Dst-e exon 23 | Dst-e exon 23 | Dst-a ex 86 | Dst-a exons 7 & 29 (2nd = Dst-e ex 19) | delete Dst-e exons 1–3, part of 4 | Dst-a ex27/ Dst-e ex14 | starts Dst-e ex 24 | Dst-e exon 23 | Dst-e exon 23 |
| shared exon Dst-a/Dst-e? | no | no | no | no | 1 yes, 1 no | yes | yes | yes | no | no |
| # of affected individuals or mouse lines reported | 1 | 7 | 1 | 3 | 1 | 1 line | 1 line | 1 line | 4 lines | 11 lines |
| carriers asymptomatic? | yes | most | most | yes | yes | yes | yes | yes | yes | yes |
| published in | Groves 2010 | Takeichi 2014 | Liu 2012 | Edvardson 2012 | Cappuccio 2017 | Guo 1995 | Kuriyama 2017 | Seehusen 2016 | this pub | this pub |
| trauma induced spontaneous blisters and erosions of lightly haired regions of skin | yes | yes | yes | no | yes | yes | yes | no | no | mild ears (pruritic?), no foot or tail |
| abnormal hair/skin in thickly haired regions | no | no | no | — | no | hair loss | no | no | no | hair loss/ blisters (blisters pruritic?) |
| age when occurred | lifelong | lifelong/ from infancy | lifelong | — | by 12 yrs | 2–4 wks old | 2–4 wks old | — | no | 8+ weeks of age |
| main locations | ankles/feet | hands, feet | soles of feet/ clothing friction points | — | soles and sites of friction | tail blistering, trunk hair loss | tail | — | no | dorsal trunk |
| blisters heal? | no delay or scarring | no delay or scarring | yes | — | yes? | retarded | NR | — | no | yes |
| hypo or hyperpigmentation? | yes | yes | no | — | yes | no | no | — | no | no |
| fingernail dystrophy? | no | no | no | — | no | no | no | — | no | no |
| toenail dystrophy? | yes, all | no | no | — | no | no | no | — | no | no |
| mucosal blistering? | no | no | no | — | no | tongue | NR | — | no | no |
| dermal-epidermal separations by light microscopy | no | NR | NR | — | NR | yes, 2–4 wks | yes, 2–4 wks | no | no | yes, >30 wks |
| hemidesmosome (HD) inner plaque (TEM) | absent or poorly formed | NR | absent or poorly formed | NR | NR | absent or poorly formed | absent or poorly formed | NR | NR | absent or poorly formed |
| intermediate filaments extend to HD? (TEM) | yes | NR | yes | NR | NR | **no** | no??? | NR | NR | **yes** |
| skin anti-Dst-e Ab or cDNA results (works on control) | BPC319 neg, 5E neg | BPC319 neg | BPC319 neg | NR | NR | 5E neg, two other Abs neg | mRNA neg | cDNA neg | cDNA pos | cDNA pos |
| skin sensory nerves by UCHL1/PGP9.5 Ab | NR | NR | NR | NR | NR | NR | **neg** | NR | NR | **pos** |
| dental involvement? | yes? | no | no | no | no | NR | NR | no | no | no? |
| caries | moderate | no | no | — | — | NR | NR | — | no | — |
| gingivitis | yes, reactive | no | no | — | — | NR | NR | — | no | — |

*(Continued)*

**Table 2.** (Continued)

| organism | human | human | human | human | human | mouse | mouse | mouse | mouse | mouse |
|---|---|---|---|---|---|---|---|---|---|---|
| erosions/misshapen teeth? | none | no | no | — | — | NR | NR | — | no | no? |
| gastrointestinal involvement? | none | no | no | maybe | yes | NR | NR | no | no | no |
| urological symptoms | none | no | no | no | NR? | NR | NR | no | no? | no |
| lung? | NR | NR | NR | NR | NR | NR | NR | suppurative inflammatory changes | | |
| neurological symptoms | mild, CADASIL, maybe Notch3 | no | none | severe/ HSAN6 | yes | dystonia musculorum | dystonia musculorum | dystonia musculorum | no | no |

NR = not reported.

Digestive tract symptoms between FSD+*jeb* with flatus and the human patient compound heterozygous for dystonin mutations with persistent diarrhea may be related [26].

Pinnae removal performed in FSD+*jeb* mice to avoid ear lesions and allow extended tail scoring had two unanticipated effects. It improved body condition (less hair loss/later lesions) and it extended 'health span'. FSD and FSD+*jeb* mice with ulcers have sometimes been observed scratching. It is therefore believed that their ear and trunk symptoms are largely pruritic, and pinnae removal causes improvement by removing the urge to scratch. Variability in histological epithelial separation in FSD+jeb mice of similar age with similar overt disease symptoms (hunching, weight loss, bloated digestive tract upon dissection) were such that attempts to correlate amount of separation with survival differences between pinnae intact and pinnae removed FSD+jeb mice were not successful. The effects of pinnae removal may be worth exploring in other mouse EB models.

## Materials & methods

### Generation, breeding and testing of dystonin (*Dst*) CRISPR/Cas9 mutated mice

Direct CRISPR/Cas9 modification of mouse zygotes was performed as described previously [46]. Briefly, microinjection of Cas9 mRNA (100ng/μl) and sgRNA (50ng/μl) targeting two locations in *Dst-e/DST-213* isoform exon 23, GCGGAAGCTCGGCGAGCCGCGG and GAGAAAGGAAAACTCCAGAGG (PAM in gray), designated as DST3A and DST3B respectively, was performed into nuclei of separate groups of C57BL/6J (B6) fertilized eggs. Each injection also included a 200nt single-stranded DNA oligo (3ng/μl) matching B6 sequence at the cut sites except mismatched at one nucleotide resulting in an amino acid change to match the 129X1 allele (1226R→Q and 1469G→V, respectively) and four synonymous SNPs for

**Table 3. 200mers.**

| Dst-213A |
|---|
| GCAGTACCGCCGTGAGCTGGAGACGATCGTGAGGGAGAAGGAAGCTGCCGAGCGGGAGCTGGAGCGGGTGAGGCAGCTCACGGCGGAGGCGGAAGCACA_A_AGGGCCGCGGTGGAGGAGAACCTCCGCAACTTCCGGAGCCAGCTGCAGGAAAACACCTTCACCAGGCAGACGCTGGAAGATCATCTGCGGAGGAAAGACT |

| Dst-213B |
|---|
| GACAAAGCTGAAGAGGTCAGGCAGGAAGCAAACGATCTCAAGAAAATAAAGCACACCTATCAGTTGGAATTAGAGTCTCTCCATCAAGAGAAGGT_C_AAACTCCAGC_GC_GAAGTTGACAGAGTCACCAGGGCGCACGCACTAGCCGAGAGGAACATTCAGTGTTTAAACTCCCAAGTTCACGCTTCGAGGGATGAGAAGGA |

Red letters indicate intended nucleotides to change. Green highlight indicates amino acid change.

genotype tracking by PCR (**Fig 1**, **Table 3**). One successful DST3A replacement founder (called *em1*, based on allele name *Dst^{em1Dcr}*) and multiple DST3A and DST3B deletion mutant founders (*em2-em13* and *em14-em16*, respectively) were identified by PCR. Each was back-crossed to B6 and to B6-*Lamc2^{jeb/jeb}* <10Mb 'short' congenic [13, 14] and made homozygous on both backgrounds (generally called IFD for *em2-5* in-frame deletions and FSD for *em6-16* frame shift deletions when homozygous on B6 and IFD+*jeb* and FSD+*jeb* when homozygous also for B6-*Lamc2^{jeb/jeb}*). Control strains used for comparison (C57BL/6J (B6), B6-*Lamc2^{jeb/jeb}*, B6-*Lamc2^{jeb/jeb} Col17a1^{PWD/PWD} Dst^{129/129}* and FVB-*Lamc2^{jeb/jeb}* were all described previously [14, 33]. *Dst* FSD lines *em13* (B6(129X1)-*Dst^{em13Dcr}*/DcrJ, JR32425) and *em14* (B6(129X1)-*Dst^{em13Dcr}*/DcrJ, JR32426) and B6-*Lamc2^{jeb/jeb}* (JR25467) are all available from JAXMice (jax.org/jax-mice-and-services). B6(129X1)-Dst<em1Dcr>/Dcr (JR29932), B6(129X1)-Dst<em5Dcr>/Dcr (JR29936) and B6(129X1)-Dst<em6Dcr>/Dcr (JR29938) were all sperm frozen without QC as private stocks and may be requested from The Jackson Laboratory. All other lines used here were not frozen and permitted to become extinct. Sanger sequencing was performed from PBL DNA PCR products to determine extent of deletions and to confirm intended DST3A replacement (**Table 1**). All procedures performed were approved by The Jackson Laboratory's IACUC.

## Phenotyping

Tail tension test and aging for ear and tail scores were performed on select genotypes and compared to appropriate controls as previously described [33, 47]. Bodies were scored similarly to ears and tails by observing the entire trunk and recording a score noting the worst hair loss and lesions on any part of it from 0 for 'unaffected' to 6 for 'very affected'. Typically scores of 1–4 denoted the amount of hair loss with no breaks in skin while scores of 5–6 were reserved for blistering/lesions. Graphed ear, tail and body score data points represent average score at age in weeks. Tension data points represent individual mice. N = 4–25 males were tension tested per line at 10 weeks of age. N = 5–27 males were ear, tail and trunk scored per line beyond 10 weeks of age until euthanasia was required due to condition, except n = 54 B6-*Lamc2^{jeb/jeb} Dst^{em14/em14}* (em14+jeb) with pinnae removed were scored, of which only 2 survived to 31 weeks of age. Due to previously documented sexual dimorphism (females have later disease onset than males), only males were tested for most comparisons. Based on previous experience, females were anticipated to have similar but delayed results [47]. B6-*Lamc2^{jeb/jeb}* (B6-*Lamc2^{jeb}*/DcrJ JR25467 'short congenic') tension values used as controls versus various *Dst em* lines are those collected during the same period. B6-*Lamc2^{jeb/jeb}* ear, tail and body scores used as controls are averages of all collected. Due to overlap of experiments, the B6-*Lamc2^{jeb/jeb}* ear and tail scores used here as controls are the same as those published in Sproule et al. 2023a Figs 1, 5, 6 and 10 [14] and Sproule et al. 2023b Figs 2 and 5 [34] as controls. Most but not all B6-*Lamc2^{jeb/jeb}* tension controls were also previously published in Sproule et al. 2023b Fig 4E [34].

## Genotyping

While single amino acid replacement required changing only one nucleotide at each target, the 200mer replacement templates included four additional synonymous SNPs in close proximity to the intended missense change to aid in genotyping and to prevent recutting of correct replacements (**Table 3**). Synonymous changes were selected to keep codon usage approximately equal and reduce likelihood they would alter phenotypes (**Fig 1D**). Replacement and wild-type specific forward primers were designed with 3' ends aligning with the intended polymorphisms. The same reverse primer was used in both replacement and wild-type reactions.

Deletions were identified using primer pairs bracketing the CRISPR cut sites at various distances to give wild-type products of 96 to 1751 bp. Various combinations of primers identified many deletions at DST3A of 19–60 bp plus larger deletions of 95, 125 and 1016 bp. Most were kept and tested but several were found to be repeats of the same 43 bp deletion and were discarded without testing. Deletions of 7, 14 and 17 bp were likewise identified, kept and tested at the DST3B cut site.

DNA was collected from mice requiring genotyping via retro-orbital blood extraction followed by washes in Buffone's Buffer and Proteinase K digestion. All primers were designed using Primer Express 2.0 software (Applied Biosystems, Inc.) using annealing temperatures of 58–60˚C. PCR used 40 cycles of 94˚C for 30 seconds, 60˚C for 60 seconds and 70˚C for 60 seconds. Band resolution was performed using 0.7% agarose, 1.5% Synergel (both from BioExpress) and 150 ng/ml Ethidium Bromide gels in TAE buffer. Primers and combinations used are listed in **Table 1**.

## Pinnae removal

Pinnae removal was performed using The Jackson Laboratory IACUC routine procedure CMQ17-03 as follows: Mice were anesthetized intraperitoneally (IP) with 2% Tribromoethanol at 20 µl/gram body weight (BW). Once at a surgical state of anesthesia, petrolatum ophthalmic ointment was applied to the eyes to prevent drying of the cornea, 50 mg/ml Carprofen analgesic at 10 µl/gram BW was injected subcutaneously (SC) and pinnae were sanitized by two applications each of 70% ethanol and 2% Chlorhexidine. Ear pinnae were held by sterile forceps away from the body and cut with sterile Iris scissors just above the cartilage crest (~2-3mm from skull). Bupivacaine topical analgesic at 1 mg/ml was applied to the site, warm 0.9% saline was injected SC at 1 ml/25 grams BW and mice were kept warm and observed until mobile. Mice were checked once per day for three days for signs of distress or complications then observed once per week for ear, tail and body scores. Ear scores for pinnae removed mice were based upon condition of the remainder of the external ear and adjacent hairless skin and only rarely exceeded '0' ('unaffected').

## Sequencing and qPCR

DNA from peripheral blood leukocytes and cDNA from tail skin were extracted from $Dst^{em}$ homozygous mice and B6 controls; segments of interest were amplified by PCR and Sanger sequencing of PCR products was performed then analyzed using Sequencher software as previously described [33]. For qPCR, primer pairs were designed and validated to give mouse $Dst$-$e$/Dst-213 ~100–200 bp cDNA products for Ex16F-17R, Ex22F-23R, Ex23F5R5 (internal to exon 23) and Ex23F-24R. cDNA was also Ex22F-24R primer tested by PCR, sequencing and qPCR and found to produce a small product indicative of exon 23 skipping in all samples, including B6 wild-type controls. For one experiment whole ear was collected as well as tail skin from two male B6 at 11 weeks old and RNA was extracted and converted to cDNA identically for all and qPCR tested with the same primers to address tissue specific expression level differences. $B2m$ or $18s$ were included in each qPCR run as a control. Purchased human adult skin cDNA (Zyagen HD-101) was similarly tested with human $DST$-$e$/DST-205 exon 22, 23 and 24 specific primers by PCR and sequencing, with human $GAPDH$ as a control. All primers used are listed in **Table 1**. N = 1 mouse tested per line in **Fig 7D** and 2 B6 in **Fig 7E**, four replicates per mouse.

## Capillary immunoelectrophoresis

Full trunk skin from euthanized 3–4 day old female mice homozygous for select $Dst$ em lines on a B6 background as well as B6 and 129X1 as controls were collected in 1.5 ml capped tubes,

snap frozen in liquid nitrogen and stored at -80°C until processed for protein. Frozen tissues were thawed, weighed and then pulverized in TT1 tissueTUBEs using the CP02 cryoPREP Impactor, (Covaris. Inc., Washburn, Mass.). Five µl of ice cold RIPA buffer, (150 mM NaCl, 1.0% IGEPAL® CA-630, 0.5% sodium deoxycholate, 0.1% SDS, 50 mM Tris, pH 8.0), was added per mg of tissue to Miltenyi M Tubes on ice. The pulverized tissues were quickly added and shaken down before being homogenized in a gentleMACS Dissociator, (Miltenyi Biotec Inc., Auburn, CA). The foam was reduced by centrifugation for 5 minutes at 2k x g before the lysates were transferred to 2 ml microtubes and centrifuged for 10 minutes at 21k x g. The supernatants were removed and each was frozen at -80°C in several aliquots. One aliquot of each mouse strain was used for protein concentration determination using the Micro BCA assay kit (Thermo Scientific, Rockford, Ill.).

Subsequent aliquots of the skin lysates were defrosted on ice and diluted to 0.4 mg/ml in 1% SDS. Dystonin quantitation for each lysate was determined using Bioworld rabbit poly-clonal antibody to dystonin, @ 30 µg/ml, (BS6535, Bioworld Technology, Saint Louis Park, MN) along with Cell Signaling rabbit mAb to a-Tubulin, 1H10 lot 11 used @ 1:20 on the Wes immunoassay system (ProteinSimple, San Jose, CA). The secondary antibody was Protein Simple ready-to-use anti-Rabbit HRP. The standard run protocol was used except the primary incubation time was extended to 120 minutes followed by three washes. N = 1 mouse tested per line; results shown are from a single run (**Fig 5A and 5B**).

## Histology and immunohistochemistry

Various combinations of dorsal trunk skin, ear pinnae, cross and longitudinal tail sections and cross sections of the foot to include footpads and nails were collected from mice at various ages in 10% neutral buffered formalin (NBF, 6 week old only) or Fekete's acid-alcohol-formalin (Fekete's) fixative for 24 hours at room temperature [48]. Feet, tail and head sections were then placed in Immunocal for 24 hours while ear and trunk skin was held in 70% ethanol. Fixed heads were cut vertically to 2–3 mm or ~6 mm cross-sections centered on eyes or full length ~5 mm sagittal sections between eyes. Samples were then embedded in paraffin blocks by the JAX Histology Service using standard protocols, 5 µm sectioned onto slides and stained with hematoxylin and eosin (H&E) or hematoxylin and an antibody directed against ubiquitin carboxy-terminal hydrolase L1 UCHL1/PGP9.5 (Sigma-Aldrich; https://tumor.informatics. jax.org/mtbwi/antibody.jsp) to identify peripheral nerves (6 week old only). Heads were step sectioned at 300 or 400 micron intervals to allow analysis at various levels. Ear and tail percent dermal-epidermal boundary separation (Figs 2J, 6H) was estimated based on observation of entire boundary length visible on H&E stained sections or UCHL1 immunohistochemistry slides. Each graphed data point represents one mouse (n = 3–12 mice per genotype at each age). Digital images used in figures here were captured with Nanozoomer (40x, 226 nm/px).

Counts of mice examined by light microscopy: n = 3 B6, 3 *em13* FSD and 3 *em14* FSD at 6 weeks old, all males; n = 8 B6, 4 B6-*Lamc2^{jeb/jeb}*, 2 *em8* FSD, 6 *em8* FSD+*jeb*, 2 *em14* FSD and 6 *em14* FSD+*jeb* at 16–18 weeks old, all males; n = 3 B6 at 26 weeks old and 3 B6, 3 *em4* IFD, 3 *em9* FSD, 3 *em13* FSD and 3 *em14* FSD (all *Lamc2^{wt/wt}*) at 30–35 weeks old, all males; n = 4 *em14* FSD at 44 weeks old and 4 *em4* IFD at 55 weeks old, all males. Tissues collected and examined: Ear and radial tail at all ages. Dorsal trunk and feet at 6 and 44–54 weeks old. Liver, lung, spleen, kidney, stomach, small intestine and large intestine at 16–18 weeks old. Muzzle to top of ribs cross sections from a subset of B6, B6-*Lamc2^{jeb/jeb}*, em8 FSD+*jeb* and em14 FSD +*jeb* at 16–18 weeks old. Additional heads cut cross-section or sagittal from B6, B6-*Lamc2^{jeb/jeb}*, em14 FSD and em14 FSD+*jeb* both with pinnae intact and pinnae removed at 16–18 weeks old.

## Measuring ear thickness

Five measurements per ear were taken at the middle of H&E stained cross sections to determine the full thickness of the ears. Measurements were taken from the top of the stratum corneum on one surface to the top of the stratum corneum of the opposite surface in areas with no tearing or other artifacts. Measurements in μm were taken using a 10x objective on an Olympus BX50 microscope (Olympus, Tokyo, Japan) using the arbitrary line measurement tool on the Olympus DP27 digital camera software.

## Transmission electron microscopy

Dorsal trunk and footpad skin samples from $CO_2$ euthanized 6 week old male *em13* and *em14* FSD homozygotes and B6 controls (n = 3 each) were collected and stored in 2% glutaraldehyde/0.1M phosphate buffer pH 7.4 at room temperature until processed. For preparing slices for observation with an electron microscope, the skin collected by the above method was postfixed with 2% glutaraldehyde/0.1M PB and 1% osmium tetroxide/0.1M PB, dehydrated with ethanol rising series and embedded in an epoxy resin. The resin-embedded skin tissue was sectioned using an Ultra Microtome (Ultracut N, Reichert-Nissei, Tokyo) to prepare an ultrathin section with a thickness of 80 nm. Ultrathin sections were double stained with uranium acetate and lead citrate and observed with a transmission electron microscope (Hitachi H 7650, Tokyo) [49, 50].

## Statistics

All statistical analysis was performed in GraphPad Prism. Tension and capillary immunoelectrophoresis statistics are 1-way ANOVA except B6 vs *em10* hom in **Fig 10** is 2-tailed t-test. Ear, tail and body score statistics are ANOVA based on age at which scores first reached a threshold of 4 ('moderately affected') for each. Storage of data in JCMS and extraction of select data sets for comparison is as previously described [14, 34]. For survival statistics both Log-rank (Mantel-Cox) and Gehan-Breslow-Wilcoxon are calculated in Prism and less significant of the two is displayed. ns = not significant, * $p<0.05$, ** $p<0.01$, *** $p<0.001$, **** $p<0.0001$.

All mouse Ensembl references are to NCBI Build 38.

## Supporting information

**S1 Table. Supporting information.**
(XLSX)

**S1 Fig. Raw image from which top part of Fig 7B was derived.**
(JPG)

**S2 Fig. Raw image from which bottom part of Fig 7B was derived.**
(JPG)

**S3 Fig. Raw image from which Fig 7F was derived.**
(JPG)

**S1 File.**
(PDF)

## Acknowledgments

We thank Dr. Ulrich Rodeck and Carla Portocarrero of Thomas Jefferson University for initially bringing the B6-*Lamc2^jeb/jeb^* wasting phenotype to our attention and JAX Microinjection,

Histology and Imaging services for excellent assistance. We also thank Drs. Hirohide Take-bayashi and Ulrike Teichmann for answering questions about their mouse models which helped direct this work.

## Author Contributions

**Conceptualization:** Thomas J. Sproule, Michael V. Wiles, Derry C. Roopenian, John P. Sundberg.

**Data curation:** Thomas J. Sproule.

**Formal analysis:** Thomas J. Sproule, Robert Y. Wilpan, Benjamin E. Low, Yudai Kabata, Rii-chiro Abe, Derry C. Roopenian, John P. Sundberg.

**Funding acquisition:** Derry C. Roopenian, John P. Sundberg.

**Investigation:** Thomas J. Sproule, Robert Y. Wilpan, John J. Wilson, Benjamin E. Low, Yudai Kabata, Riichiro Abe, John P. Sundberg.

**Methodology:** Thomas J. Sproule.

**Project administration:** Derry C. Roopenian, John P. Sundberg.

**Supervision:** Derry C. Roopenian.

**Visualization:** Thomas J. Sproule.

**Writing – original draft:** Thomas J. Sproule, Derry C. Roopenian, John P. Sundberg.

**Writing – review & editing:** Thomas J. Sproule, Robert Y. Wilpan, John J. Wilson, Benjamin E. Low, Yudai Kabata, Tatsuo Ushiki, Riichiro Abe, Michael V. Wiles, Derry C. Roopenian, John P. Sundberg.

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
