## [Decision Letter · Decision Letter 0]

29 Aug 2023

PONE-D-23-22545Dystonin modifiers of junctional epidermolysis bullosa and models of epidermolysis bullosa simplex without dystonia musculorumPLOS ONE

Dear Dr.Sproule,

Thank you for submitting your manuscript to PLOS ONE. After careful consideration, we feel that it has merit but does not fully meet PLOS ONE’s publication criteria as it currently stands. Therefore, we invite you to submit a revised version of the manuscript that addresses the points raised during the review process.

Although neither one of the reviewers suggested additional experiments, reviewer 2 felt - and I agree - that the manuscript could benefit from streamlining including shortening of the text and restructuring of data presentation. Several additional points raised need clarification and consideration.

Please submit your revised manuscript within 3 wweks. If you will need more time than this to complete your revisions, please reply to this message or contact the journal office at plosone@plos.org. Please include the following items when submitting your revised manuscript:A rebuttal letter that responds to each point raised by the academic editor and reviewer(s). You should upload this letter as a separate file labeled 'Response to Reviewers'.A marked-up copy of your manuscript that highlights changes made to the original version. You should upload this as a separate file labeled 'Revised Manuscript with Track Changes'.An unmarked version of your revised paper without tracked changes. You should upload this as a separate file labeled 'Manuscript'.

We look forward to receiving your revised manuscript.

Kind regards,

Gerhard Wiche, Ph.D.

Academic Editor

PLOS ONE

Journal Requirements:

2. We noted in your submission details that a portion of your manuscript may have been presented or published elsewhere. [The current draft of this Manuscript includes the following statement: "Due to overlap of experiments, the B6-Lamc2jeb/jeb ear and tail scores used here as controls are the same as those published in Sproule et al. 2023a figures 1, 5, 6 and 10 [12] and Sproule et al. 2023b figures 2 and 5 [45] as controls . Most but not all B6-Lamc2jeb/jeb tension controls were also previously published in Sproule et al. 2023b figure 4E [45]."] Please clarify whether this [conference proceeding or publication] was peer-reviewed and formally published. If this work was previously peer-reviewed and published, in the cover letter please provide the reason that this work does not constitute dual publication and should be included in the current manuscript.

Reviewers' comments:

Reviewer's Responses to Questions

**Comments to the Author**

1. Is the manuscript technically sound, and do the data support the conclusions?

Reviewer #1: Yes

Reviewer #2: Yes

2. Has the statistical analysis been performed appropriately and rigorously? 

Reviewer #1: Yes

Reviewer #2: Yes

3. Have the authors made all data underlying the findings in their manuscript fully available?

Reviewer #1: Yes

Reviewer #2: Yes

4. Is the manuscript presented in an intelligible fashion and written in standard English?

Reviewer #1: Yes

Reviewer #2: Yes

5. Review Comments to the Author

Reviewer #1: This paper extends the authors' efforts to clarify the genetic modifications affecting epidermolysis bullosa (EB) using mouse models. The authors focus on Dst and demonstrate that mutations in Dst exon 23 can protect the junctional EB (JEB) phenotype. Through their analyses, the authors rediscover the rodless Dst isoform and find that removal of the pinnae can ameliorate EB skin phenotypes in mice. The study is well-conducted and informative. I have some comments as follows:

1) In some figures, B6 is used to represent Lamc2wt/wt, but in others, B6 stands for Lamc2jeb/jeb. Consistency in nomenclature across all figures would be preferable.

2) The rediscovery of the rodless Dst1 isoform is intriguing. Please emphasize that the removal of exon 23 is in-frame, enabling readers to more readily understand that exon 23 skipping does not lead to nonsense-mediated mRNA decay.

3) Is it possible that pinnae removal could improve inflammatory phenotypes observed in other EB model mice (e.g., Lama3 targeted disruption; PMID 27729280)? If so, please make a note of it in the manuscript.

Reviewer #2: This study presents mouse models of dystonin-associated epidermolysis bullosa simplex and also from these models implicates dystonin as a modifier of laminin-332-associated junctional epidermolysis bullosa. It is a well-conducted, comprehensive work. The presented findings are highly interesting and will be of importance for our understanding of dystonin biology, epidermolysis bullosa and the models will be useful for therapy research.

The manuscript is very rich in data and this makes it a quite challenging read. In addition, some findings are limitedly supported due to few mice or biological replicates analyzed. In my opinion the work would greatly benefit from restructuring the manuscript to focus on the key findings and findings of more supportive nature or other interesting observations deserve to be kept but I think that they could just be shortly mentioned and the data and more detailed explanation of them presented as supporting information. 12 figures are to much for the reader to effectively take in.

In the introduction it is stated in regard to modifiers of disease in epidermolysis bullosa (lines 46-48) that no additional modifiers apart from MMP1 have been identified. This statement is in my opinion too strong. There has been associations in patients made of higher decorin expression and milder diseases in recessive dystrophic epidermolysis bullosa (Odorisio et al., Human Mol Genet 2014), a similar link was made for PRELP (Chacon-Solano et al., Matrix Biol 2022).

The correct abbreviation for laminin-332 is LM332.

The observations of pinnae removal delaying development of severe phenotypes and extending healthy span are interesting. The suggestion that is secondary due to reduced scratching of ears is reasonable. However, did the authors also consider autoimmunity?

The discussion is interesting but I think it is too long to be effective. I would recommend to shorten it an focus on discussing essential findings.

6. PLOS authors have the option to publish the peer review history of their article (what does this mean?). If published, this will include your full peer review and any attached files.

Reviewer #1: No

Reviewer #2: No

---

## [Author Response · Author response to Decision Letter 0]

2 Oct 2023

Thank you for your thoughtful comments concerning our submitted manuscript PONE-D-23-22545 Dystonin modifiers of junctional epidermolysis bullosa and models of epidermolysis bullosa simplex without dystonia musculorum.

Editor comments.

“Although neither one of the reviewers suggested additional experiments, reviewer 2 felt - and I agree - that the manuscript could benefit from streamlining including shortening of the text and restructuring of data presentation. Several additional points raised need clarification and consideration.”

Response: We have made another pass through the manuscript resulting in some clarification and modest streamlining. 

Reviewer 1 comments.

“In some figures, B6 is used to represent Lamc2wt/wt, but in others, B6 stands for Lamc2jeb/jeb. Consistency in nomenclature across all figures would be preferable.”

Response: Figures 3 and 10 have been changed to have B6-jeb stand for Lamc2jeb/jeb.

“The rediscovery of the rodless Dst1 isoform is intriguing. Please emphasize that the removal of exon 23 is in-frame, enabling readers to more readily understand that exon 23 skipping does not lead to nonsense-mediated mRNA decay.”

Response: Changes were made to emphasize this. 

“Is it possible that pinnae removal could improve inflammatory phenotypes observed in other EB model mice (e.g., Lama3 targeted disruption; PMID 27729280)? If so, please make a note of it in the manuscript.”

Response: We added a sentence to the discussion suggesting it may be worth exploring pinnae removal in other mouse EB models.

Reviewer 2 comments.

“In the introduction it is stated in regard to modifiers of disease in epidermolysis bullosa (lines 46-48) that no additional modifiers apart from MMP1 have been identified. This statement is in my opinion too strong. There has been associations in patients made of higher decorin expression and milder diseases in recessive dystrophic epidermolysis bullosa (Odorisio et al., Human Mol Genet 2014), a similar link was made for PRELP (Chacon-Solano et al., Matrix Biol 2022).”

Response: This statement has been modified and both of the citations mentioned here added to references.

“The correct abbreviation for laminin-332 is LM332.”

Response: Actually this statement is not correct. We checked with Michelle Perry, PhD, of Mouse Genome Informatics at The Jackson Laboratory. She is the contact person for all questions on mouse genetic nomenclature. She stated “We don’t have official nomenclature guidelines for protein complexes. The more common version (at least from Google hits) appears to be Ln332 rather than L332 or LM332.” Therefore, we decided to leave it as L332 to match our previous publication.

“The observations of pinnae removal delaying development of severe phenotypes and extending healthy span are interesting. The suggestion that is secondary due to reduced scratching of ears is reasonable. However, did the authors also consider autoimmunity?”

Response: Autoimmunity is one of many interesting possible explanations. The lab has closed down and we no longer have the option to test any of these. We only mention scratching as a ‘most likely’ possible explanation.

“The discussion is interesting but I think it is too long to be effective. I would recommend to shorten it an focus on discussing essential findings.”

Response: We have made efforts to clarify and condense where we could, resulting in modest shortening of the text. We hope this is sufficient to satisfy the reviewer and editor.

---

## [Decision Letter · Decision Letter 1]

9 Oct 2023

Dystonin modifiers of junctional epidermolysis bullosa and models of epidermolysis bullosa simplex without dystonia musculorum

PONE-D-23-22545R1

Dear Dr. Sproule,

We’re pleased to inform you that your manuscript has been judged scientifically suitable for publication. One of the reviewers found the manuscript could still be further streamlined but there was no opposition to acceptance. The manuscript will be formally accepted for publication once it meets all outstanding technical requirements.

Kind regards,

Gerhard Wiche, Ph.D.

Academic Editor

PLOS ONE

Additional Editor Comments (optional):

Reviewers' comments:

Reviewer's Responses to Questions

**Comments to the Author**

1. If the authors have adequately addressed your comments raised in a previous round of review and you feel that this manuscript is now acceptable for publication, you may indicate that here to bypass the “Comments to the Author” section, enter your conflict of interest statement in the “Confidential to Editor” section, and submit your "Accept" recommendation.

Reviewer #1: All comments have been addressed

Reviewer #2: (No Response)

2. Is the manuscript technically sound, and do the data support the conclusions?

Reviewer #1: Yes

Reviewer #2: Yes

3. Has the statistical analysis been performed appropriately and rigorously? 

Reviewer #1: Yes

Reviewer #2: Yes

4. Have the authors made all data underlying the findings in their manuscript fully available?

Reviewer #1: Yes

Reviewer #2: Yes

5. Is the manuscript presented in an intelligible fashion and written in standard English?

Reviewer #1: Yes

Reviewer #2: Yes

6. Review Comments to the Author

Reviewer #1: (No Response)

Reviewer #2: Thank you for revising the manuscript and considering my suggestions.

The preferred abbreviation for laminin is LM (Aumailley et al., 2005 - A simplified laminin nomenclature doi: 10.1016/j.matbio.2005.05.006). However, you are of course free to use any abbreviation you like. This is not the reason to why I am recommending another round of revisions. I think that the findings are highly interesting, however, there is still a lack of streamlining and prioritization of data which make the manuscript challenging to read. 12 main figures are not optimal for a study like this. I would highly recommend you to perform additional streamlining of the manuscript.

7. PLOS authors have the option to publish the peer review history of their article (what does this mean?). If published, this will include your full peer review and any attached files.

Reviewer #1: No

Reviewer #2: No

---

## [Editor Report · Acceptance letter]

16 Oct 2023

PONE-D-23-22545R1 

Dystonin modifiers of junctional epidermolysis bullosa and models of epidermolysis bullosa simplex without dystonia musculorumCan the post-ruminal urea release impact liver metabolism, and nutritional status of beef cows at late gestation? 

Dear Dr. Sproule:

I'm pleased to inform you that your manuscript has been deemed suitable for publication in PLOS ONE. Congratulations! Your manuscript is now with our production department. 

Kind regards, 

on behalf of

Prof. Gerhard Wiche 

Academic Editor

PLOS ONE